# Cryo-EM structures of an LRRC8 chimera with native functional properties reveal heptameric assembly

**Hirohide Takahashi[1,2], Toshiki Yamada[3], Jerod S Denton[3,4], Kevin Strange[3], Erkan Karakas[1,2]***

[1]Department of Molecular Physiology and Biophysics, School of Medicine, Vanderbilt University, Nashville, United States; [2]Center for Structural Biology, Vanderbilt University, Nashville, United States; [3]Department of Anesthesiology, Vanderbilt University Medical Center, Nashville, United States; [4]Department of Pharmacology, School of Medicine, Vanderbilt University, Nashville, United States

**Abstract** Volume-regulated anion channels (VRACs) mediate volume regulatory Cl⁻ and organic solute efflux from vertebrate cells. VRACs are heteromeric assemblies of LRRC8A-E proteins with unknown stoichiometries. Homomeric LRRC8A and LRRC8D channels have a small pore, hexameric structure. However, these channels are either non-functional or exhibit abnormal regulation and pharmacology, limiting their utility for structure-function analyses. We circumvented these limitations by developing novel homomeric LRRC8 chimeric channels with functional properties consistent with those of native VRAC/LRRC8 channels. We demonstrate here that the LRRC8C-LRRC8A(IL1²⁵) chimera comprising LRRC8C and 25 amino acids unique to the first intracellular loop (IL1) of LRRC8A has a heptameric structure like that of homologous pannexin channels. Unlike homomeric LRRC8A and LRRC8D channels, heptameric LRRC8C-LRRC8A(IL1²⁵) channels have a large-diameter pore similar to that estimated for native VRACs, exhibit normal DCPIB pharmacology, and have higher permeability to large organic anions. Lipid-like densities are located between LRRC8C-LRRC8A(IL1²⁵) subunits and occlude the channel pore. Our findings provide new insights into VRAC/LRRC8 channel structure and suggest that lipids may play important roles in channel gating and regulation.

*For correspondence: erkan.karakas@vanderbilt.edu

## Editor's evaluation

This paper is valuable to the field of ion channels, as it provides useful information about the assembly of the volume-regulated anions channels formed by LRRC8 proteins. The evidence that the LRRC8C-LRRC8A chimera has native functional properties and adopts a heptameric assembly is convincing. The evidence supporting the physiological relevance of the heptameric assembly and the hypothesized role of lipids is incomplete and will be addressed in future studies.

## Introduction

Volume-regulated anion channels (VRACs) are expressed widely in vertebrate cell types where they mediate the efflux of Cl⁻ and organic solutes required for cell volume regulation (**Jentsch, 2016**; **Strange et al., 2019**). VRACs are activated by increases in cell volume and by large reductions in intracellular ionic strength (**Strange et al., 2019**).

VRACs are encoded by the *Lrrc8* gene family (**Qiu et al., 2014**; **Voss et al., 2014**), which comprises five paralogs termed *Lrrc8a-e* (**Abascal and Zardoya, 2012**; **Voss et al., 2014**). Native VRAC/LRRC8 channels are heteromers with unknown stoichiometry. LRRC8A is an essential VRAC/LRRC8 subunit

**Figure 1.** Construct design of 8C-8A(IL1²⁵). (**A**) Sequence alignment of LRRC8A and LRRC8C around the swapped IL1²⁵ region. The swapped region in the 8C-8A(IL1²⁵) construct is shown inside a black box. The key below amino acid sequences denotes identical (*), conservative (:), and semi-conservative (.) sequences. Space indicates the residues that are not conserved. (**B**) The amino acid sequence of the 8C-8A(IL1²⁵) construct around the swapped IL1²⁵ region. (**C**) Schematic diagram of the 8C-8A(IL1²⁵) protein highlighting the relative position of the swapped IL1²⁵ region.

The online version of this article includes the following source data and figure supplement(s) for figure 1:

**Figure supplement 1.** Purification of 8C-8A(IL1²⁵).

**Figure supplement 1—source data 1.** Raw and annotated images of the SDS-PAGE gel analyzing the purified LRRC8A and 8C-8A(IL1²⁵) proteins.

**Figure supplement 1—source data 2.** Raw and annotated images of the native-PAGE gel analyzing the purified LRRC8A and 8C-8A(IL1²⁵) proteins.

and must be co-expressed with at least one other paralog to reconstitute volume-regulated channel activity (*Syeda et al., 2016*; *Voss et al., 2014*). High-resolution cryo-electron microscopy (cryo-EM) structures have been determined for homomeric LRRC8A (*Deneka et al., 2021*; *Deneka et al., 2018*; *Kasuya et al., 2018*; *Kefauver et al., 2018*; *Kern et al., 2019*) and LRRC8D (*Nakamura et al., 2020*) channels demonstrating that they have a hexameric configuration.

Defining the molecular basis by which VRAC and other volume-sensitive channels detect cell volume changes is a fundamental and longstanding physiological problem. Detailed molecular understanding of this important problem requires accurate channel structural information. Homomeric LRRC8A and LRRC8D channels have abnormal functional properties or are not expressed in the plasma membrane (*Voss et al., 2014*; *Yamada et al., 2021*; *Yamada and Strange, 2018*; *Zhou et al., 2018*). Structure-function studies of VRAC/LRRC8 heteromeric channels are complicated by their undefined, likely variable, and experimentally uncontrollable stoichiometry. To circumvent these problems, we developed a series of novel homomeric LRRC8 chimeras that exhibit functional properties similar to the native heteromeric VRAC/LRRC8 channels (*Yamada and Strange, 2018*).

We describe here the cryo-EM structures of the LRRC8C-LRRC8A(IL1²⁵) chimera, hereafter termed 8C-8A(IL1²⁵). 8C-8A(IL1²⁵) consists of a 25-amino acid sequence unique to the first intracellular loop, IL1, of LRRC8A inserted into the corresponding region of LRRC8C (*Figure 1*). Like native VRAC/LRRC8 channels, 8C-8A(IL1²⁵) chimeras are activated strongly by cell swelling and low intracellular ionic strength (*Yamada and Strange, 2018*). We demonstrate that the 8C-8A(IL1²⁵) chimeric channel

is a large-pore seven-subunit heptamer, like homologous pannexin channels (*Bhat and Sajjad, 2021*; *Deng et al., 2020*; *Jin et al., 2020*; *Kuzuya et al., 2022*; *Michalski et al., 2020*; *Mou et al., 2020*; *Qu et al., 2020*; *Ruan et al., 2020*). The pore diameter is similar to that estimated for native VRACs, and permeability to large organic anions is significantly higher than that of homohexameric LRRC8A channels. Lipid-like densities are located between 8C-8A(IL1$^{25}$) channel subunits and occlude the channel pore, as has been shown recently for human pannexin 1 (*Kuzuya et al., 2022*). Our results, together with the previous studies (*Deneka et al., 2021*; *Deneka et al., 2018*; *Kasuya et al., 2018*; *Kefauver et al., 2018*; *Kern et al., 2019*; *Nakamura et al., 2020*), demonstrate that LRRC8 proteins can form anion channels with different subunit numbers.

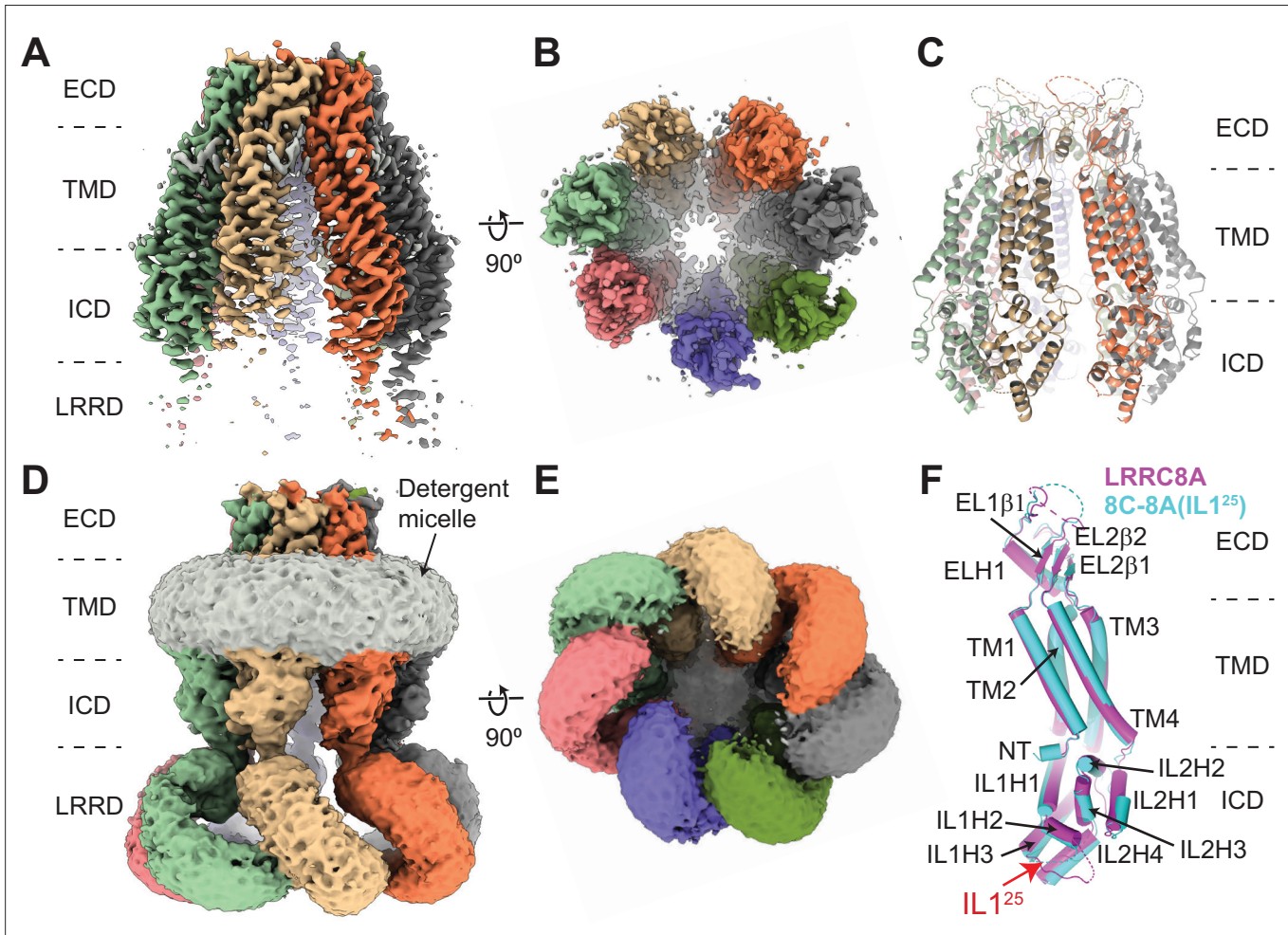

**Figure 2.** Cryo-electron microscopy (cryo-EM) structure of 8C-8A(IL1$^{25}$). (**A–B**) Cryo-EM maps of 8C-8A(IL1$^{25}$) class 1 structure viewed through the membrane plane (**A**) and from the cytoplasm (**B**). (**C**) Ribbon representation of the 8C-8A(IL1$^{25}$) class 1 structure viewed through the membrane plane. (**D–E**) Unsharpened cryo-EM maps of 8C-8A(IL1$^{25}$) class 1 structure viewed through the membrane plane (**D**) and from the cytoplasm (**E**), emphasizing low-resolution features. (**F**) Structural comparison of the 8C-8A(IL1$^{25}$) (cyan) and LRRC8A (magenta, PDB ID: 5ZSU) subunits.

The online version of this article includes the following figure supplement(s) for figure 2:

**Figure supplement 1.** Cryo-electron microscopy (cryo-EM) analysis of 8C-8A(IL1$^{25}$).

**Figure supplement 2.** Cryo-electron microscopy (cryo-EM) analysis of 8C-8A(IL1$^{25}$).

**Figure supplement 3.** Structural comparison of 8C-8A(IL1$^{25}$) to LRRC8A and pannexins.

## Results

### Structure determination

To facilitate structure-function understanding of VRAC/LRRC8 channel regulation and pore properties, we expressed the 8C-8A(IL1[25]) chimera in Sf9 insect cells and purified the detergent-solubilized complexes by affinity and size exclusion chromatography (*Figure 1—figure supplement 1*). The purified 8C-8A(IL1[25]) complex appeared larger than the LRRC8A hexamers in both native-PAGE and size exclusion chromatography analysis (*Figure 1—figure supplement 1B–C*). We performed single-particle cryo-EM analysis to determine the structure of the complex (*Figure 2* and *Figure 2—figure supplements 1 and 2*). Following 2D classification and ab initio 3D reconstruction, we performed 3D

**Table 1.** Cryo-electron microscopy (cryo-EM) data collection, refinement, and validation statistics.

**Data collection and processing**

| | | | | | |
|---|---|---|---|---|---|
| Microscope | FEI Krios G3i microscope | | | | |
| Detector | Gatan K3 direct electron camera | | | | |
| Nominal magnification | ×81,000 | | | | |
| Voltage (kV) | 300 | | | | |
| Electron exposure (e/Å²) | 54 | | | | |
| Defocus range (μm) | −0.8 to −1.5 | | | | |
| Pixel size (Å) | 1.1 | | | | |
| Number of Micrographs | 3198 | | | | |
| Particles images (no.) | 846,122 | | | | |
| Conformational state | Class 1 | Class 2 | Class 3 | Class 4 | Class 5 |
| Symmetry imposed | C1 | C1 | C1 | C1 | C1 |
| Final particles images (no.) | 203,011 | 132,722 | 100,772 | 93,179 | 85,591 |
| Map resolution (Å) (FSC threshold = 0.143) | 3.4 | 3.6 | 3.7 | 3.8 | 4.0 |
| **Refinement** | | | | | |
| Model resolution (Å) (original map, FSC threshold = 0.5) | 3.6 | 3.9 | 3.9 | 4.2 | 4.4 |
| B-factor used for map sharpening (Å²) | −102.0 | −88.2 | −84.7 | −70.5 | −83 |
| *Model composition* | | | | | |
| Non-hydrogen atoms | 17,499 | 17,499 | 17,465 | 10,435 | 10,435 |
| Protein residues | 2101 | 2101 | 2097 | 2101 | 2101 |
| *Mean B factors (Å²)* | | | | | |
| Protein | 32.4 | 23.5 | 32.4 | 44.9 | 38.34 |
| *R.m.s. deviations* | | | | | |
| Bond lengths (Å) | 0.003 | 0.002 | 0.003 | 0.002 | 0.004 |
| Bond angles (°) | 0.563 | 0.525 | 0.553 | 0.516 | 0.807 |
| *Molprobity score* | 1.72 | 1.68 | 1.75 | 1.37 | 1.49 |
| *Clash score* | 5.04 | 4.84 | 6.02 | 1.14 | 1.54 |
| *Poor rotamers (%)* | 0.0 | 0.0 | 0.0 | 0.0 | 0.0 |
| Favored (%) | 92.9 | 93.5 | 93.6 | 89.95 | 88.3 |
| Allowed (%) | 7.1 | 6.5 | 6.4 | 10.0 | 11.6 |
| Disallowed (%) | 0 | 0 | 0 | 0.05 | 0.1 |

classification and obtained five distinct 3D classes for 8C-8A(IL1$^{25}$) (*Figure 2—figure supplement 1*). The 8C-8A(IL1$^{25}$) maps showed no apparent symmetric arrangement. Therefore, the final reconstructions were done without enforcing any symmetry and resulted in five structures with resolutions in the range of 3.4–4.0 Å (*Figure 2A and B*, *Figure 2—figure supplements 1 and 2*, and *Table 1*).

Similar to LRRC8A and LRRC8D structures, the 8C-8A(IL1$^{25}$) structure comprises four domains, the extracellular domain (ECD), transmembrane domain (TMD), intracellular domain (ICD), and leucine-rich repeat (LRR) motif-containing domain (LRRD) (*Figure 2*). The cryo-EM maps for the ECD and TMD revealed high-resolution features allowing us to build an atomic model comprising most of the ECD and TMD except residues 60–94 in the ECD and the first 15 residues of the N-terminus. The quality of the cryo-EM maps for the ICDs was improved by performing local refinements, and the resulting maps were used to build a model (*Figure 2—figure supplement 2*). However, the first intracellular loop containing the swapped IL1$^{25}$ region remained unresolved. Although the unsharpened maps revealed clear features for the entire protein, the local resolution for the LRRD was insufficient to build an atomic model, and the local refinement strategy we applied did not provide any meaningful improvement in LRRD resolution (*Figure 2A–E* and *Figure 2—figure supplements 1 and 2*). Therefore, we did not build an atomic model for the LRRD and used the maps without B-factor sharpening to assess their structure and overall arrangement relative to the rest of the protein complex (*Figure 2D–E*).

## 8C-8A(IL1$^{25}$) chimeras form heptameric channels

The overall structure of an individual 8C-8A(IL1$^{25}$) subunit is similar to that of LRRC8A (*Figure 2F*) and LRRC8D. However, unlike homomeric LRRC8A and LRRC8D channels, which are hexamers (*Deneka et al., 2021*; *Deneka et al., 2018*; *Kasuya et al., 2018*; *Kefauver et al., 2018*; *Kern et al., 2019*; *Nakamura et al., 2020*), the 8C-8A(IL1$^{25}$) chimeric channel is a seven-subunit heptamer, similar to the homologous pannexin channels (*Figure 2B and E*; *Figure 2—figure supplement 3*; *Bhat and Sajjad, 2021*; *Deng et al., 2020*; *Jin et al., 2020*; *Kuzuya et al., 2022*; *Michalski et al., 2020*; *Mou et al., 2020*; *Qu et al., 2020*; *Ruan et al., 2020*).

The subunit arrangement of the 8C-8A(IL1$^{25}$) heptameric channel is asymmetric as opposed to LRRC8A and LRRC8D hexamers, which have two-, three-, or sixfold symmetric arrangements (*Deneka et al., 2021*; *Deneka et al., 2018*; *Kasuya et al., 2018*; *Kefauver et al., 2018*; *Kern et al., 2019*; *Nakamura et al., 2020*). When the ICD and TMD are viewed from the cytoplasm, 8C-8A(IL1$^{25}$) subunits are organized into two distinct groups, one with four subunits and the other with three subunits (*Figure 3A–D*). These two groups of subunits associate via a loose interface, while the subunits within each group associate via a tight interface (*Figure 3A–D* and *Figure 3—figure supplements 1–3*).

In contrast to the ICDs and TMDs, the ECDs are arranged symmetrically and form extensive contacts between the subunits in both tight and loose interfaces (*Figure 3E* and *Figure 3—figure supplement 1*). Most of the subunit interaction is between the residues on ELH1 and the linker that connects EL2β1 and EL2β2 in one subunit and the residues on the three β-strands of the neighboring subunit (*Figure 3E* and *Figure 3—figure supplement 1*). The contacts between each subunit pair bury about 1900 Å$^2$ of solvent-accessible surface area at the ECDs (*Table 2*). The buried solvent-accessible surface areas between the subunit pairs in the TMDs and ICDs are considerably smaller and exhibit high variability between the tight and the loose interfaces. In the TMD, the buried areas are about 650 and 110 Å$^2$ for the tight and loose interfaces, respectively (*Table 2*). The residues within the core of the TMD are loosely associated and the major interactions of the subunits within the TMD are mediated by the residues closer to the ECDs or ICDs at the tight interface (*Figure 3—figure supplement 2*). At the loose interface, only residues closer to the ECD are in close range for direct interaction (*Figure 3—figure supplement 2*). In the ICD, the buried areas in the tight interfaces range from 290 to 560 Å$^2$, whereas there are no solvent-inaccessible contacts at the loose interfaces (*Figure 3E*, *Figure 3—figure supplements 2–3*, and *Table 2*).

The asymmetric arrangement observed for the TMDs and ICDs is likely due to their divergent orientation relative to the ECDs. *Figure 4A* shows the alignment of the seven subunits on their ECDs. TMDs do not align and exhibit up to an 8° difference in their orientation relative to the ECD, while the difference is larger for the ICDs (*Figure 4A*). Another notable difference between the subunits is in the linker that connects the first β-strand on extracellular loop 1, EL1β1, of the ECD to the transmembrane helix 1 (TM1) of the TMD (*Figure 4A–B*). In two of the subunits in the class 1 structure, K51 points toward the pore, while D50 points in the opposite direction. In the five other subunits,

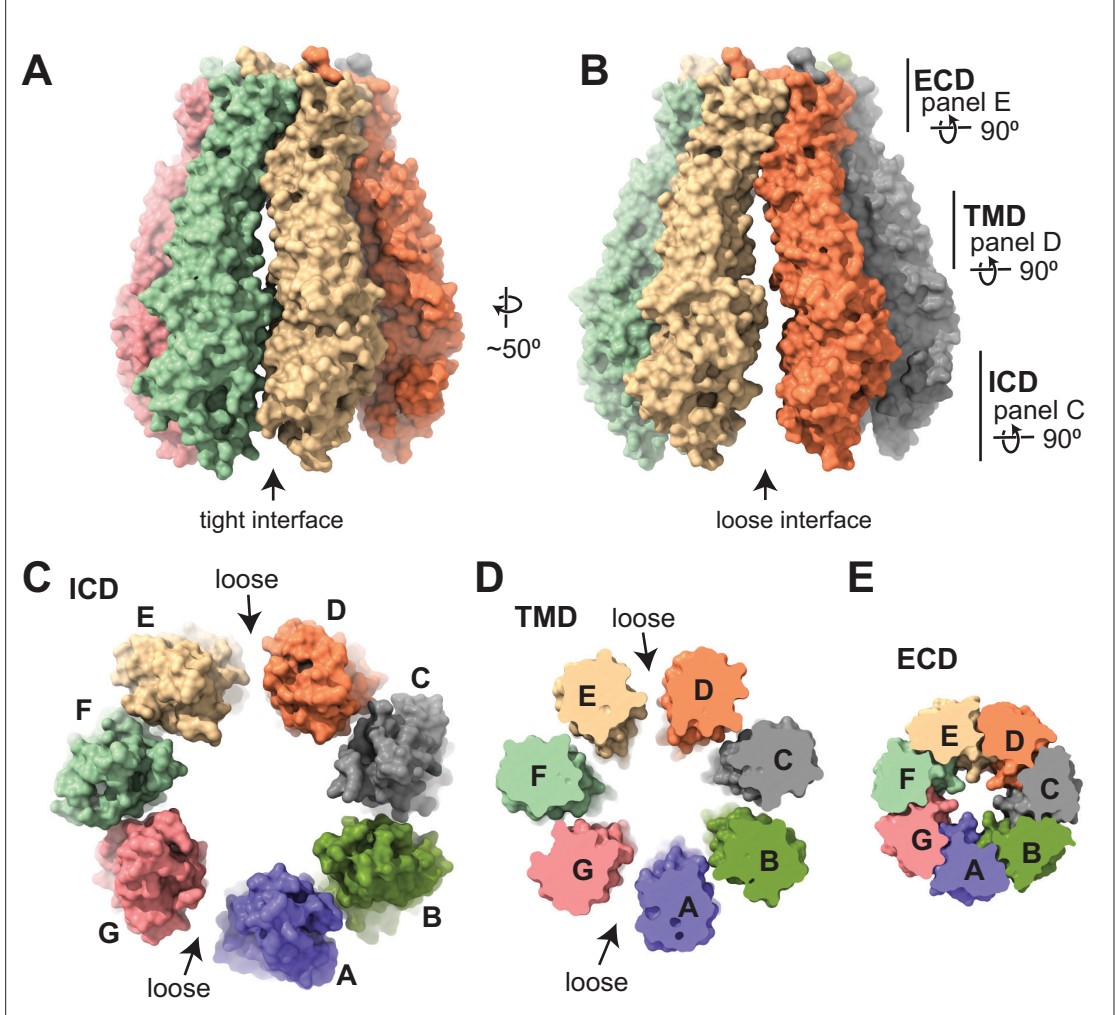

**Figure 3.** Subunit arrangement of the 8C-8A(IL1$^{25}$) chimera. (**A–B**) Surface representation of the 8C-8A(IL1$^{25}$) class 1 structure viewed from two sides, highlighting the 'tight' (**A**) and 'loose' (**B**) interfaces. (**C–E**) Intracellular domain (ICD), transmembrane domain (TMD), and extracellular domains (ECDs) are viewed from the cytoplasm with a depth of view, as shown in panel (**B**).

The online version of this article includes the following figure supplement(s) for figure 3:

**Figure supplement 1.** Comparison of subunit interfaces.

**Figure supplement 2.** Comparison of subunit interfaces.

**Figure supplement 3.** Comparison of subunit interfaces.

D50 points toward the pore, and K51 is oriented toward the subunit interface (*Figure 4C*). When K51 points toward the pore, it creates a groove between the subunits (*Figure 4D*). These grooves, located between the subunits forming the loose interface in the class 1 structure, connect the space within the ECD to the membrane-facing surface of the TMD (*Figure 4E*). When D50 points toward the pore, the side chain of K51 occupies this space, closing the groove (*Figure 4F*).

## Structural heterogeneity of 8C-8A(IL1$^{25}$) chimeras

We obtained five distinct 3D classes for the 8C-8A(IL1$^{25}$) chimera. The most apparent difference between the structures is in the arrangement of the LRRDs (*Figure 5A–B*). In class 1, all seven LRRDs are arranged circularly with roughly sevenfold symmetry. For class 5, one of the LRRDs is positioned outside of the quaternary assembly formed by six LRRDs arranged with pseudo-twofold symmetry. The density for the LRRD located outside the assembly is poorly visible, indicating high flexibility relative to the rest of the complex (*Figure 5A–B*). The arrangements of the LRRDs in the other three 3D classes exhibit diverse arrangements and subunit interactions. As a result, several different subunit

**Table 2.** Total buried solvent-accessible surface area between subunits for each domain.

| | Buried surface area between the neighboring subunits (Å²)* | | | | | | |
|---|---|---|---|---|---|---|---|
| | Subunits A-B | Subunits B-C | Subunits C-D | Subunits D-E | Subunits E-F | Subunits F-G | Subunits G-A |
| | Tight | Tight | Tight | **Loose** | Tight | Tight | **Loose** |
| ECD[†] | 1995 | 1897 | 1964 | **1905** | 1900 | 1896 | **1869** |
| TMD[†] | 664 | 625 | 620 | **122** | 660 | 670 | **99** |
| ICD[†] | 462 | 294 | 500 | **0** | 559 | 502 | **0** |

*Buried solvent-accessible surface area calculations were performed using the software NACCESS v2.1.1 (**Hubbard and Thornton, 1993**).

[†]Domain definitions used for these calculations are as follows: ECD: residues 49–121 and 288–310; TMD: residues 20–48, 122–150, 260–287, and 311–342; ICD: residues 151–259 and 343–405.

interfaces exist between the neighboring LRRDs. However, a detailed analysis of these interfaces is not possible due to the limited resolution of the cryo-EM maps for these regions, prohibiting the building of the models with amino acid assignments.

Consistent with the conformational differences in the arrangements of the LRRDs, the ICDs also exhibit conformational heterogeneity, albeit less pronounced, among the five 3D classes we observed (*Figure 5C* and *Figure 5—figure supplement 1*). *Figure 5C* shows the arrangement of the ICDs and the distances between the Cα atoms of G163 of the neighboring subunits. The non-symmetric and diverse subunit arrangement is evident. The distances between the subunits forming the tight interfaces appear similar in all classes, whereas the distances between the subunits forming the loose interfaces are variable among different classes. A similar variation is also observed within the TMD (*Figure 5D* and *Figure 5—figure supplement 1*). However, there are no apparent structural differences at the ECD of the structures (*Figure 5—figure supplement 1*).

The distinct arrangement of the loops formed by D50 and K51 is also observed in other classes (*Figure 5—figure supplement 2*). However, the number of subunits with K51 pointing toward the pores differs. For example, the class 2 structure has only one subunit with K51 pointing toward the pore, compared to two subunits in the class 1 structure (*Figure 5—figure supplement 2*). The groove created by the distinct K51 arrangement is located between the subunits that form one of the two loose interfaces in the class 2 structure, despite the presence of two loose interfaces. It is plausible that this loop is flexible and adopts distinct conformations, possibly depending on the orientation of the TMD relative to the ECD. However, it is not clear if there is any direct correlation between the K51 orientation and the loose interfaces.

## Pore structure

*Figure 6* shows the pore domain of the 8C-8A(IL1[25]) heptameric chimera compared to the pore domain of LRRC8A hexameric channels. The narrowest region of the heptameric 8C-8A(IL1[25]) channel pore has an average solvent-accessible radius of 4.7 Å and is formed by L105 located on the extracellular side of the protein (*Figure 6A*). This is considerably larger than the pore radius of 2.0 Å determined from cryo-EM structures of hexameric LRRC8A channels (*Figure 6B–C*; *Deneka et al., 2018*; *Kasuya et al., 2018*; *Kefauver et al., 2018*; *Kern et al., 2019*).

The difference in pore diameters of hexameric LRRC8A channels and heptameric 8C-8A(IL1[25]) channels implies that subunit number impacts channel solute permeability. To assess this directly, we quantified the relative permeabilities (i.e., $P_x/P_{Cl}$) of LRRC8A and 8C-8A(IL1[25]) to the large organic anions glutathione and lactobionate. $P_{glutathione}/P_{Cl}$ and $P_{lactobionate}/P_{Cl}$ were both significantly (p<0.01) higher for the 8C-8A(IL1[25]) heptameric channel (*Figure 6D–E*), consistent with its larger pore diameter.

## Interaction of lipids with the 8C-8A(IL1[25])

As shown in *Figure 7A*, we observe non-protein densities penetrating through the openings between the subunits in the TMD. The shape and length of these densities are consistent with lipid molecules.

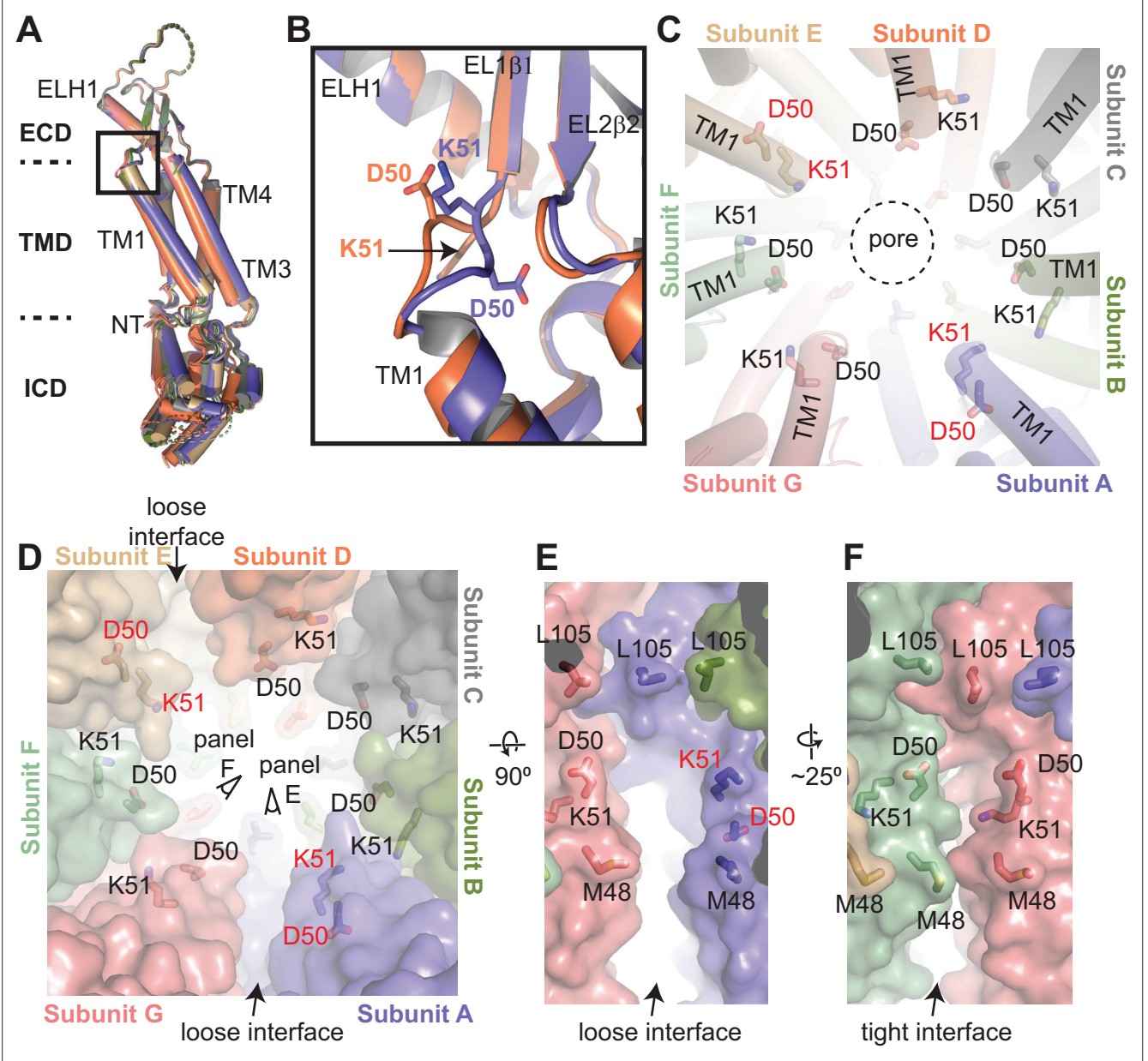

**Figure 4.** Structural heterogeneity of 8C-8A(IL1[25]) protomers. (**A**) Structural comparison of the 8C-8A(IL1[25]) subunits in the class 1 structure. The structures are aligned based on their extracellular domains (ECDs). (**B**) Close-up view of the box region in panel (**A**), highlighting the structural differences in the loop that connects transmembrane helix 1 (TM1) to EL1β1. Only two subunits are shown. (**C**) Close-up view of the pore around the residues D50 and K51, which are shown as sticks. The residues that adopt different conformations compared to others are labeled in red. The dashed circle indicates the pore-lining surface. (**D**) The same view as panel (C) but including the surface representation (transparent) to highlight the distinct interfaces between the subunits. (**E–F**) Close-up view of the interfaces from the points of view shown in panel (**D**).

Similar densities are observed in LRRC8A hexameric channels reconstituted in lipid nanodiscs (**Kern et al., 2019**). We observe these lipid-like densities on both the cytoplasmic and extracellular sides of the TMD. The lipid-like densities on the extracellular side do not show apparent differences between tight and loose interfaces. However, the lipid-like density closer to the intracellular side is considerably weaker at the loose interfaces compared to the ones at the tight interfaces (**Figure 7A**). This difference may be due to the larger separation of the subunits at the loose interface, creating more room for the lipid molecules to move.

Although the quality of the maps does not allow us to identify the lipid molecules leading to the densities we observe, the ones with the strongest features suggest phospholipids as the most likely

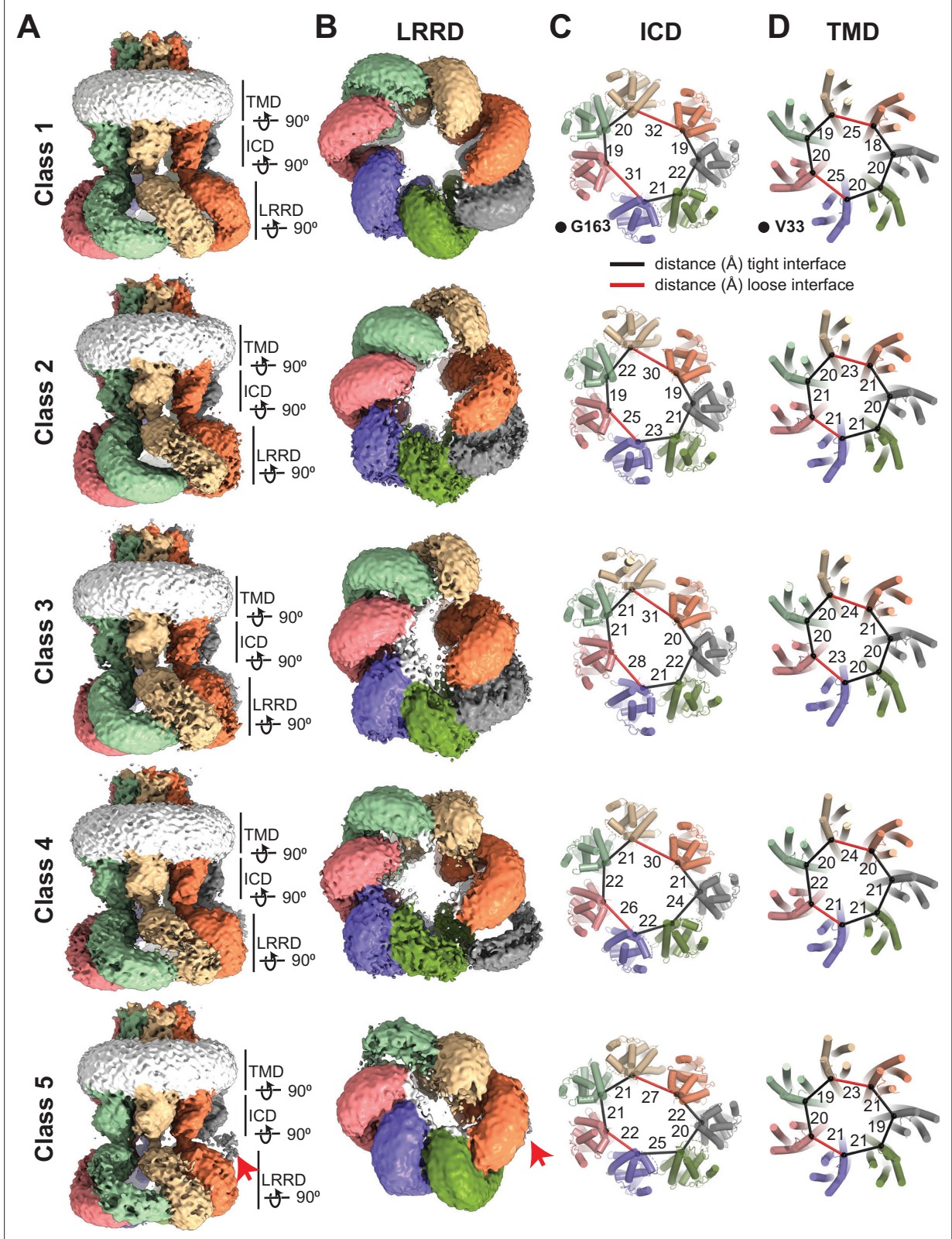

**Figure 5.** Conformational heterogeneity of 8C-8A(IL1[25]) chimeras. (**A–B**) Cryo-electron microscopy (cryo-EM) maps (before sharpening) of 8C-8A(IL1[25]) 3D classes viewed through the membrane plane (**A**) and from the cytoplasm (**B**). Individual subunits are colored as in *Figure 1*. The detergent micelle is shown in white. Red arrows point to the weak density of the leucine-rich repeat motif-containing domain (LRRD), which is not located within the LRRD quaternary assembly. Lines represent the depth of the view for LRRD, intracellular domain (ICD), and transmembrane domain (TMD). (**C–D**) Ribbon

*Figure 5 continued on next page*

*Figure 5 continued*

representation of the ICD (**C**) and TMD (**D**) structures viewed from the cytoplasm with a depth of view indicated in panel (A). The Cα atoms for G163 in ICD and V33 in TMD are shown as spheres. The distances between these atoms in neighboring subunits are shown as lines (in black and red for the tight and loose interfaces, respectively) and labeled in Å.

The online version of this article includes the following figure supplement(s) for figure 5:

**Figure supplement 1.** Comparison of 8C-8A(IL1²⁵) structures.

**Figure supplement 2.** Comparison of the D40-K51 loop among different classes.

candidates (*Figure 7B*). As shown in *Figure 7B*, a phospholipid molecule can be reasonably fitted into the strongest density we observe. The polar phosphate moiety of the lipid molecule is coordinated by two lysine residues (K125 and K318) and a histidine residue (H314) of the neighboring subunits (*Figure 7B*). W122, Y126, and L130 are close to the acyl chains (*Figure 7B*). One of the acyl chains extends through the space between the subunit and reaches the interior surface of the channel. The second acyl chain is less defined. The area contains additional non-protein densities suggesting the presence of other ordered lipid molecules.

When we examined the surface properties of the pore-lining residues, we found high densities of charged amino acid residues that align with the phospholipid head groups of the membrane bilayer surrounding the TMD (*Figure 7C*). A hydrophobic patch of amino acid residues aligns with the hydrophobic core of the membrane bilayer (*Figure 7D*).

Intriguingly, we observed two layers of non-protein densities within the channel pore that align with the boundaries of the hydrophobic patch and the detergent micelle around the TMD in the cryo-EM maps for all classes (*Figure 7E*). Similar densities have been observed in the structures of other large-pore channels, including pannexins and innexin (*Burendei et al., 2020*; *Kuzuya et al., 2022*). It is plausible that these densities are of a lipid bilayer that occupies the pore as illustrated in *Figure 7F*. In this model, the hydrophobic acyl chains would occupy the space encircled by the hydrophobic patch, and the polar head groups would interact with the charged residues at the borders of the hydrophobic patch.

To confirm that these densities are not detergent-induced artifacts, we reconstituted the 8C-8A(IL1²⁵) chimera in lipid nanodiscs and performed cryo-EM analysis (*Figure 7—figure supplement 1*). Although the resolution of the cryo-EM maps could not be improved beyond 7 Å, the same bilayer-like density within the channel pore was visible (*Figure 7—figure supplement 1*). It is noteworthy that the lipid-like densities within the pores are similar in intensity and shape to the lipid densities observed around the protein within the nanodisc (*Figure 7—figure supplement 1*).

## Discussion

Except for LRRC8A, homomeric LRRC8 channels do not traffic to the plasma membrane and cannot be functionally characterized (*Voss et al., 2014*; *Yamada and Strange, 2018*; *Zhou et al., 2018*). LRRC8A homomers form plasma membrane channels, but they exhibit abnormal functional properties. Most notably, LRRC8A channels are not activated by cell swelling under normal physiological conditions (*Yamada et al., 2021*; *Yamada and Strange, 2018*) and are only weakly activated by extreme reductions in cytoplasmic ionic strength (*Figure 6—figure supplement 1*; *Yamada et al., 2021*). LRRC8A channels also exhibit grossly abnormal DCPIB pharmacology (*Yamada et al., 2021*). The non-native functional properties of LRRC8A channels indicate that they have a non-native structure, which presents significant limitations for understanding how VRAC/LRRC8 channels are regulated and how pore structure determines solute permeability characteristics.

Homomeric LRRC8 channels with physiologically relevant functional properties are formed by chimeric proteins containing parts of the essential subunit LRRC8A and another LRRC8 paralog (*Yamada and Strange, 2018*). The 8C-8A(IL1²⁵) chimeric channel exhibits normal sensitivity to cell volume changes and intracellular ionic strength and normal DCPIB pharmacology (*Figure 6—figure supplement 1*; *Yamada et al., 2021*; *Yamada and Strange, 2018*).

Unlike homomeric LRRC8A and LRRC8D channels, which are hexamers, our studies demonstrate that 8C-8A(IL1²⁵) chimeric channels have a heptameric structure (*Figures 2–3*). The number of subunits required to form native VRAC/LRRC8 channels is unknown. However, native gel, crosslinking,

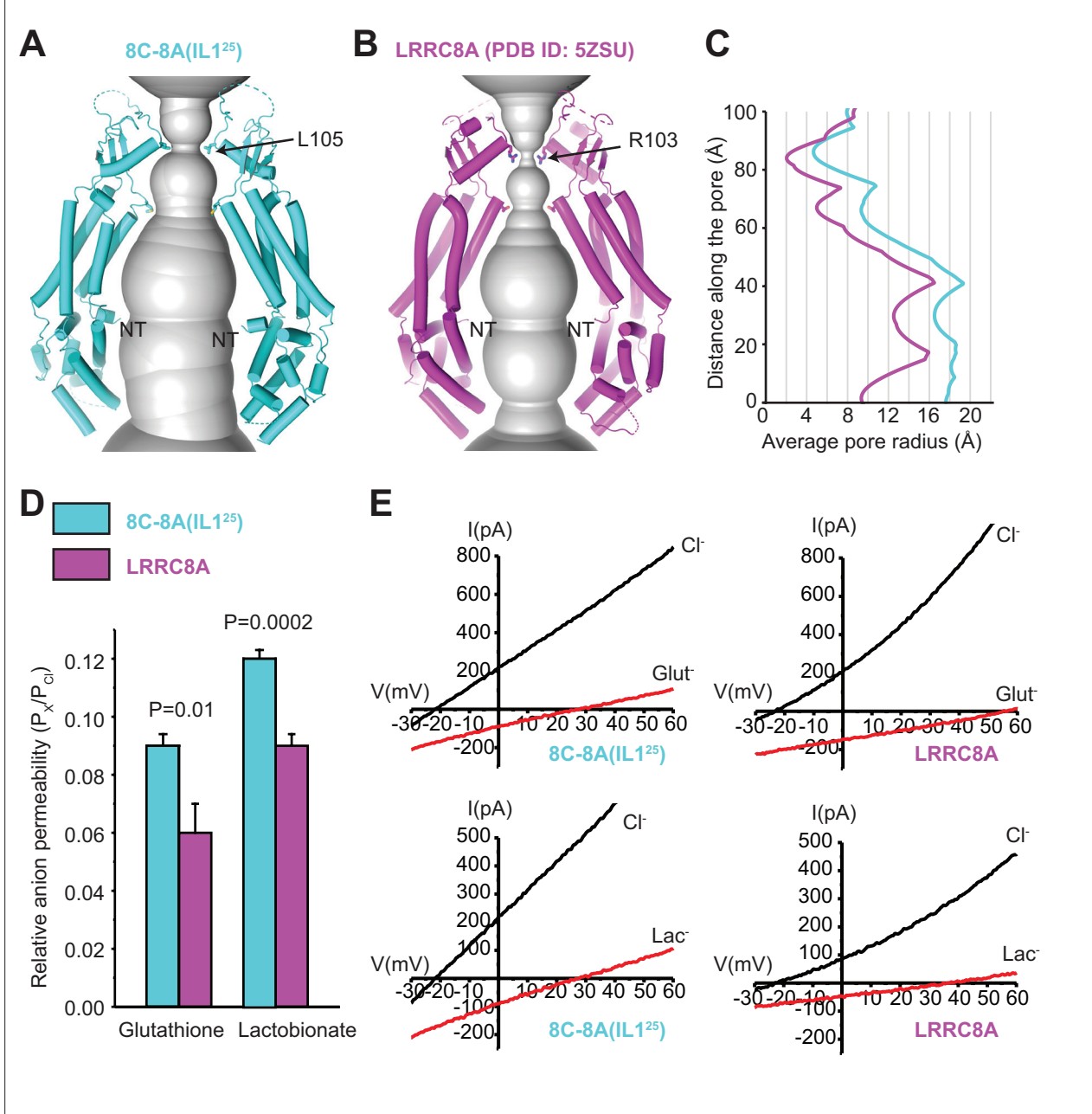

**Figure 6.** Comparison of channel pores. (**A–B**) Pore openings of the 8C-8A(IL1$^{25}$) heptameric channel (class 1 structure) (**A**) and LRRC8A (PDB ID: 5ZSU) homohexameric channel (**B**) calculated using the software program HOLE (*Smart et al., 1996*). Only two opposing subunits are shown. Residues forming the constriction sites are shown as sticks. The first modeled residues at the N-termini are labeled as NT. (**C**) 1D graph of the average radius along the length of the 8C-8A(IL1$^{25}$) (cyan) and LRRC8A (magenta) channel pores. (**D**) Relative ($P_x/P_{Cl}$) glutathione and lactobionate permeabilities calculated from reversal potential changes induced by replacing bath Cl$^-$ with the test anion. Values are means ± SEM (N=4–7). Statistical significance was determined using Student's unpaired *t-test*. (**E**) Representative LRRC8A and 8C-8A(IL1$^{25}$) current traces in the presence of bath Cl$^-$ or after substitution with glutathione (Glut$^-$) or lactobionate (Lac$^-$). Currents were elicited by ramping membrane voltage from −100 to +100 mV.

The online version of this article includes the following figure supplement(s) for figure 6:

**Figure supplement 1.** Effect of intracellular ionic strength on activation of LRRC8A and the 8C-8A(IL1$^{25}$) chimeric channel.

**Figure supplement 2.** Heptameric oligomerization leads to larger pore sizes.

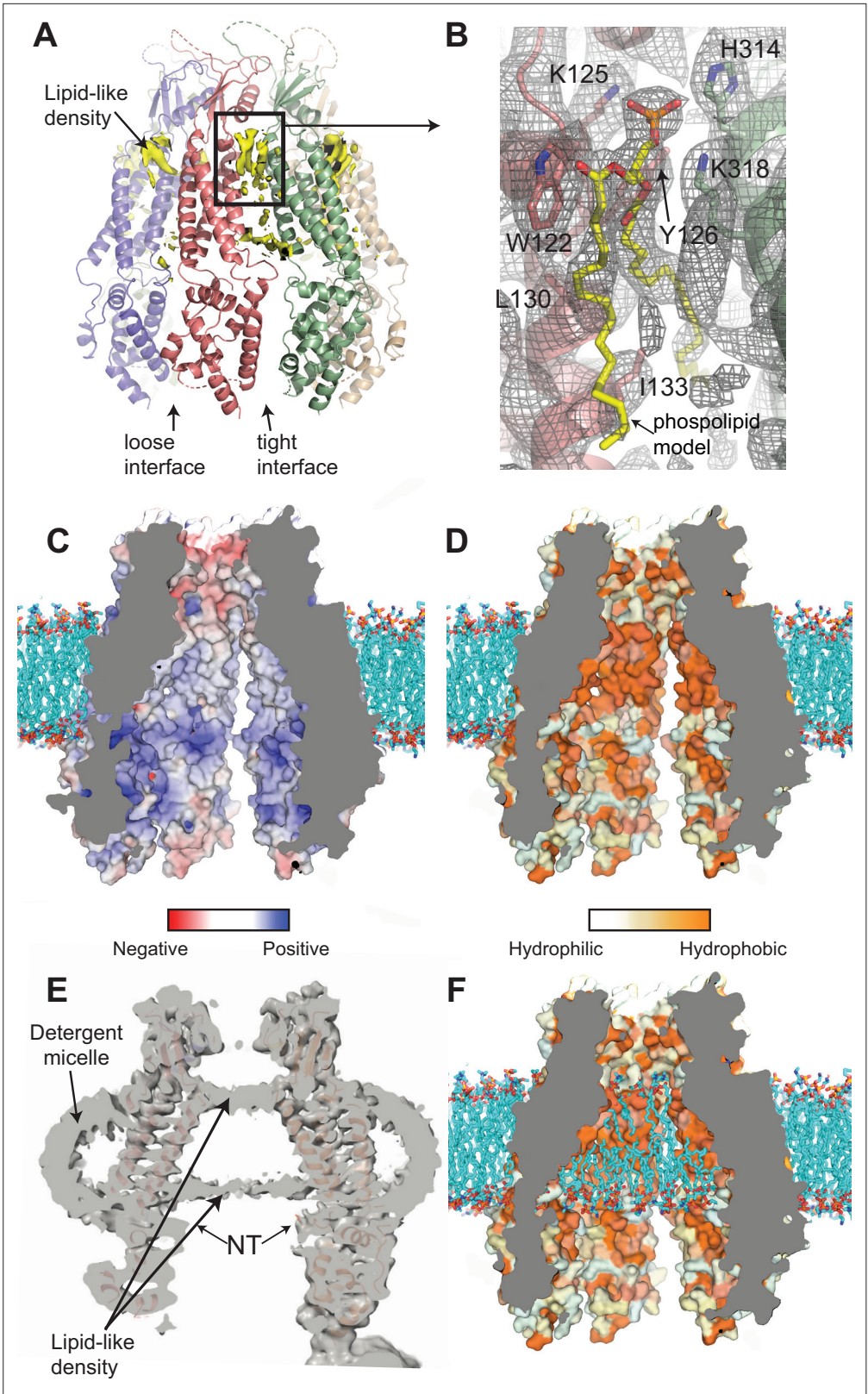

**Figure 7.** Interaction of lipids with the 8C-8A(IL1[25]) chimera. (**A**) Ribbon representation of the 8C-8A(IL1[25]) class 1 structure along with the lipid-like cryo-electron microscopy (cryo-EM) densities (yellow) between the subunits viewed through the membrane plane. Loose and tight interfaces are indicated with arrows. (**B**) Close-up views of the boxed regions in panel (A). The cryo-EM map for the entire region visualized is shown as gray mesh.

*Figure 7 continued on next page*

*Figure 7 continued*

The phospholipid molecule shown as yellow sticks within the lipid-like density is used for illustration purposes only and not included in the deposited coordinate files. Select residues near the lipid-like densities are shown as sticks and labeled. (**C–D**) Surface representation of the 8C-8A(IL1$^{25}$) pore colored based on electrostatic charge (**C**) and hydrophobicity (**D**). The hypothetical lipid bilayer around the transmembrane domain (TMD) is built using CHARMM-GUI (**Wu et al., 2014**) and shown as sticks (cyan). Positioning of the protein within the bilayer is calculated using PPM 2.0 (**Lomize et al., 2012**). (**E**) A sliced view of the unsharpened cryo-EM map (gray, transparent) with the ribbon representation of the 8C-8A(IL1$^{25}$) class 1 structure. Arrows indicate densities corresponding to pore-blocking lipid-like densities and detergent micelles. (**F**) Hypothetical interaction of lipids with the inner surface of the pore is shown by placing phospholipids using CHARMM-GUI (**Wu et al., 2014**). Surface representation of the 8C-8A(IL1$^{25}$) colored based on hydrophobicity as in panel (D).

The online version of this article includes the following figure supplement(s) for figure 7:

**Figure supplement 1.** Cryo-electron microscopy (cryo-EM) analysis of 8C-8A(IL1$^{25}$) reconstituted in nanodiscs.

and mass spectrometry studies (**Syeda et al., 2016**) and photobleaching experiments (**Gaitán-Peñas et al., 2016**) suggest that VRAC/LRRC8 channels are formed by ≥6 LRRC8 subunits.

The 8C-8A(IL1$^{25}$) has a limiting pore radius of 4.7 Å (**Figure 6** and **Figure 6—figure supplement 2A**), which unlike the much narrower LRRC8A pore (**Figure 6** and **Figure 6—figure supplement 2B**), is very consistent with the 6–7 Å pore radius estimated for native VRAC/LRRC8 channels (**Droogmans et al., 1999**; **Ternovsky et al., 2004**) and the role of these channels play in the transport of large anionic and uncharged organic solutes (**Sabirov and Okada, 2005**; **Strange et al., 2019**). A leucine residue located at position 105 forms the 8C-8A(IL1$^{25}$) pore constriction (**Figure 6A**). In LRRC8A and LRRC8B, L105 is replaced by arginine. LRRC8D and LRRC8E paralogs have either phenylalanine or leucine at the homologous position. Hexameric LRRC8A and LRRC8D channels have limiting pore radii of 2.0 and 3.5 Å, respectively (**Figure 6B** and **Figure 6—figure supplement 2**). When we create hypothetical heptamers by aligning LRRC8A and LRRC8D subunits onto individual 8C-8A(IL1$^{25}$) subunits, the pore radii increase to 4.7 and 7.1 Å, respectively, indicating that the oligomeric state has a direct impact in pore size and likely in channel transport properties (**Figure 6—figure supplement 2D–F**). Consistent with this conclusion, LRRC8A hexameric channels have lower relative permeability to the large organic anions glutathione and lactobionate compared to heptameric 8C-8A(IL1$^{25}$) channels (**Figure 6D–E**).

The narrow pore radius of 2 Å of the LRRC8A hexameric channel (**Figure 6B**) also likely accounts for its abnormal DCPIB pharmacology. Native VRAC/LRRC8 channels are inhibited >90% by 10 μM DCPIB. Inhibition is voltage-insensitive and exhibits a Hill coefficient of 2.9 (**Friard et al., 2017**), suggesting that multiple DCPIB molecules are required to inhibit the channel. The DCPIB pharmacology of 8C-8A(IL1$^{25}$) recapitulates that of native VRAC/LRRC8 channels (**Yamada et al., 2021**). In contrast, LRRC8A hexameric channels are weakly inhibited by 10 μM DCPIB, and inhibition is strongly voltage-dependent with a Hill coefficient close to 1 (**Yamada et al., 2021**). The low Hill coefficient is consistent with an LRRC8A cryo-EM structure described by **Kern et al., 2019**, showing a single DCPIB molecule blocking the channel pore at the narrowest constriction formed by R103 (**Figure 6B**).

Two recent studies have defined high-resolution cryo-EM structures of LRRC8A/LRRC8C heteromeric channels (**Kern et al., 2022**; **Rutz et al., 2023**). Both studies demonstrated that the heteromeric channels could adopt a hexameric conformation with a limiting pore radius of 2–3 Å, similar to that of LRRC8A hexamers. However, these channels either had 5:1 (**Kern et al., 2022**) or 4:2 LRRC8A:LRRC8C stoichiometries, indicating that LRRC8A:LRRC8C heteromers can adopt multiple oligomeric forms.

Taken together, existing cryo-EM structural data suggest that native VRAC/LRRC8 channels may exist in multiple oligomeric conformations, as has been reported for CALHM channels. It will be critical to relate the functional properties of various channel oligomeric conformations to those of native VRAC/LRRC8 channels. It will also be important to define the role of LRRC8A in channel assembly and conformation. Four groups have shown that LRRC8A has a dominant-negative function. Overexpression of LRRC8A suppresses endogenous (**Qiu et al., 2014**; **Syeda et al., 2016**; **Voss et al., 2014**) and heterologously expressed (**Yamada et al., 2016**) VRAC/LRRC8 currents. LRRC8A protein levels may therefore impact channel assembly and function.

Kern et al. identified lipids located between individual subunits in LRRC8A hexameric channels reconstituted in lipid nanodiscs (**Kern et al., 2019**). Intersubunit non-protein densities, which are

most likely lipids, are also observed between 8C-8A(IL1$^{25}$) subunits in detergent reconstituted channels (*Figure 7A*). Importantly, we observed two layers of density resembling lipid bilayer blocking the pore of the 8C-8A(IL1$^{25}$) heptameric channel (*Figure 7* and *Figure 7—figure supplement 1*). In agreement with our observations, in a recent pre-print reporting the hexameric structure of LRRC8A-LRRC8C (*Kern et al., 2022*), ordered lipid molecules are observed on the extracellular side of the pore opening aligning well with the densities we observed. Intrapore lipids have recently been proposed to be critical structural and regulatory elements of pannexins (*Kuzuya et al., 2022*) and closely related innexin (*Burendei et al., 2020*) and CALHM (*Drożdżyk et al., 2020*; *Syrjanen et al., 2020*) channels, as well as bacterial mechanosensitive channels (*Rasmussen et al., 2019*; *Reddy et al., 2019*; *Zhang et al., 2021*). Kuzuya et al. suggest that movement of lipids out of the pore through gaps between the subunits of the human pannexin 1 channel is required for channel opening (*Kuzuya et al., 2022*). A similar mechanism can be suggested for 8C-8A(IL1$^{25}$) heptameric channels. However, further experimental characterization is required to assess if lipids occupy the pore and play an essential role in channel gating.

Leucine-rich repeat motifs have been identified in over 14,000 proteins from viruses to eukaryotes and play essential roles in signal transduction and as sites for protein-protein interactions (*Bella et al., 2008*; *Kobe and Kajava, 2001*; *Matsushima et al., 2019*). The LRR motif can also function as a mechanosensor (*Ju et al., 2016*; *Ju et al., 2015*). VRAC/LRRC8 channels most likely arose by fusion of the transmembrane domain encoding portion of a pannexin channel gene with the LRRD portion of an unrelated gene type (*Abascal and Zardoya, 2012*). Conformational changes in the VRAC/LRRC8 channel LRR domain are correlated with changes in channel activity (*Deneka et al., 2021*; *König et al., 2019*), and multiple cryo-EM structures demonstrate that this region is conformationally flexible (*Figure 5*; *Deneka et al., 2021*; *Deneka et al., 2018*; *Kasuya et al., 2018*; *Kefauver et al., 2018*; *Kern et al., 2019*; *Nakamura et al., 2020*). These data all point to the high likelihood that the VRAC/LRRC8 channel LRRD is a critical element of the channel's cell volume sensing apparatus.

LRRC8 chimeras with normal cell volume and ionic strength sensitivity must contain all or part of the LRRC8A IL1 (*Yamada and Strange, 2018*), indicating that this protein region is also critical for channel regulation. The 25-amino acid region of the LRRC8A IL1 inserted into the 8C-8A(IL1$^{25}$) chimera is predicted to be intrinsically disordered (*Yamada and Strange, 2018*). Consistent with this prediction, this region was not visible in our cryo-EM structures. It is conceivable that the LRRC8A IL1 is required for correct channel assembly and conformation, that it plays a direct role in cell volume sensing and/or that it functions to transduce cell volume-induced conformational changes into changes in the gating.

The mechanisms by which eukaryotic cells sense cell volume changes and transduce those changes into regulatory responses remain mysterious. Understanding how VRAC/LRRC8 channels detect cell volume is greatly constrained by the abnormal functional properties of LRRC8A homomers and by the undefined heteromeric nature of native channels. The high-resolution cryo-EM structures of the 8C-8A(IL1$^{25}$) homomeric chimera, which has native physiological properties, have revealed several novel structural elements of VRAC/LRRC8 channels. These structures now provide the foundation for defining the molecular basis of channel cell volume sensing utilizing structure-guided mutagenesis combined with electrophysiological functional analysis of channels with defined stoichiometry and subunit arrangement.

## Methods

### Constructs

Human LRRC8A and LRRC8C cDNAs cloned into pCMV6 were purchased from OriGene Technologies. The 8C-8A(IL1$^{25}$) chimera cDNA construct was generated using the Phusion High-Fidelity PCR kit (New England BioLabs). All cDNAs were tagged on their carboxy terminus with Myc-DDK epitopes. For protein expression, the cDNAs encoding human LRRC8A and the 8C-8A(IL1$^{25}$) chimera were subcloned with a C-terminal FLAG tag into pAceBac1 vectors and incorporated into baculovirus using the Multibac expression system (*Fitzgerald et al., 2006*). All constructs were verified by DNA sequencing.

### Cell lines

Patch-clamp experiments were performed using *Lrrc8*$^{-/-}$ HCT116 cell line in which the five *Lrrc8* genes were disrupted by genome editing. *Lrrc8*$^{-/-}$ cells were prepared and authenticated in the Thomas

Jentsch Lab (*Ullrich et al., 2016*). They were a generous gift from Thomas Jentsch. The absence of VRAC/LRRC8 channel activity in the *Lrrc8*[-/-] cell line was confirmed routinely by patch-clamp electrophysiology performed on cells transfected with GFP cDNA only. The cells were tested negative for *Mycoplasma* contamination.

The cells were grown in McCoy's 5A media (HyClone) supplemented with 10% fetal bovine serum (R&D Systems), 50 units/ml of penicillin, and 50 µg/ml streptomycin in a humidified incubator at 37°C with 5% $CO_2$.

## Protein expression and purification

8C-8A(IL1[25]) chimera was expressed in *Sf9* cells ($4 \times 10^6$ cells/ml) at 27°C for 48 hr. Cells were harvested by centrifugation ($2000 \times g$) and resuspended in a lysis buffer composed of 150 mM NaCl and 50 mM Tris-HCl, pH 8.0, 1 mM phenylmethylsulfonyl fluoride. After cell lysis using Avastin EmulsiFlex-C3, the cell lysate was clarified from large debris by centrifugation at $6000 \times g$ for 20 min. The cleared lysate was centrifuged at $185,000 \times g$ (Type Ti45 rotor, Beckman) for 1 hr. Membrane pellets were resuspended and homogenized in ice-cold resuspension buffer (150 mM NaCl, 50 mM Tris-HCl, pH 8.0) and solubilized using 0.5% lauryl maltose neopentyl glycol (LMNG) at a membrane concentration of 100 mg/ml. The solubilized pellets were stirred gently for 4 hr at 4°C, and the insoluble material was separated by centrifugation at $185,000 \times g$ (Type Ti45 rotor, Beckman) for 40 min. The supernatant was then mixed with anti-FLAG affinity gel resin (Sigma) at 4°C for 1 hr. After washing the resin with 10 column volume wash buffer composed of 150 mM NaCl, 50 mM Tris-HCl, pH 8.0, and 0.005% LMNG, the protein was eluted using the wash buffer supplemented with 100 µg/ml FLAG peptide. Protein was further purified by size exclusion chromatography using a Superose 6 Increase column (10/300 GL, GE Healthcare) equilibrated with 150 mM NaCl, 50 mM Tris-HCl, pH 8.0, 0.005% LMNG. The peak fraction corresponding to 8C-8A(IL1[25]) was concentrated to 3.0 mg/ml, centrifuged at $220,000 \times g$ using an S110-AT rotor (Thermo Fisher Scientific) for 10 min, and used immediately for cryo-EM imaging. Human LRRC8A with a C-terminal Flag tag was purified using the same protocol described above for 8C-8A(IL1[25]).

For nanodisc incorporation, 8C-8A(IL1[25]) chimera was purified as described above except *n*-dodecyl-β-D-maltopyranoside (DDM, 1% for solubilization of the membrane and 0.05% in the purification buffers) was used instead of LMNG. The membrane scaffold protein MSP1E3D1 was expressed using the p MSP1E3D1 plasmid, a gift from Stephen Sligar (Addgene plasmid # 20066; http://n2t.net/addgene:20066; RRID: Addgene_20066) (*Denisov et al., 2007*). MSP1E3D1 was expressed in *Escherichia coli* BL21(DE3) cells and purified as described previously using $Ni^{2+}$ affinity resin (*Ritchie et al., 2009*). The histidine tag was cleaved off using TEV protease, and the protein was further purified by SEC using a HiLoad 16/600 Superdex 200 pg column equilibrated with 300 mM NaCl and 40 mM Tris-HCl, pH 8.0. Peak fractions containing MSP1E3D1 were collected and stored at –80°C for the nanodisc formation.

## MSP1E3D1 nanodisc formation

The preparation of nanodiscs was performed by mixing 8C-8A(IL1[25]) purified in the presence of DDM with POPC lipids (Avanti Polar Lipids, Inc) and MSP1E3D1 at a final molar ratio of 1:2.5:250 (8C-8A(IL1[25]):MSP1E3D1:POPC). The mixture was incubated with Biobeads SM2 (Bio-Rad) overnight. After removing the biobeads by centrifugation, the protein sample was concentrated and purified by SEC using a Superose 6 10/300 Increase column equilibrated with 150 mM NaCl and 50 mM Tris-HCl, pH 8.0. Peak fractions corresponding to 8C-8A(IL1[25])-MSP1E3D1 nanodiscs were concentrated to 2.0 mg/ml and used for cryo-EM grid preparation immediately.

## Cryo-EM sample preparation and data collection

Purified 8C-8A(IL1[25]) was applied to 300 mesh UltrAuFoil holey gold 1.2/1.3 grids (Quantifoil Microtools) that were glow discharged for 10 s at 25 mA. The grids were blotted for 4 s at force 12 using double-layer Whatman filter papers (1442-005, GE Healthcare) before plunging into liquid ethane using an FEI MarkIV Vitrobot at 8°C and 100% humidity. Samples were imaged using a 300 kV FEI Krios G3i microscope equipped with a Gatan K3 direct electron camera. Movies containing 400 frames were collected in super-resolution mode at ×81,000 magnification with a physical pixel size of 1.1 Å/

pixel and defocus values at a range of –0.8 to –1.5 μm using the automated imaging software SerialEM (*Mastronarde, 2005*).

8C-8A(IL1$^{25}$) nanodisc grids were prepared as described above. Samples were imaged using a 300 kV FEI Krios G4 microscope equipped with a Gatan K3 direct electron camera. Movies containing 50 frames were collected in super-resolution mode at ×103,000 magnification with a physical pixel size of 0.818 Å/pixel and defocus values at a range of –0.8 to –2.2 μm using the automated imaging software EPU (Thermo Fisher Scientific).

## Cryo-EM data processing

All image processing was performed using CryoSparc2 (*Punjani et al., 2017*). Motion correction and CTF estimations were performed locally using Patch Motion Correction and Patch CTF Estimation procedures. Initial particle picking was performed by blob search. Particles were then binned 4× and extracted. After 2D classification, classes with clear structural features were selected and used for template-based particle picking. The new set of particles was binned four times and extracted. After 2D classification, a cleaned particle set was used for the ab initio 3D reconstruction. The resulting map was used as a model for 3D classification. After a series of 2D and 3D classification runs, particles were reextracted with a box size of 360×360 pixels. These particles were separated into six classes using 3D classification. Five out of six classes revealed interpretable density maps, and these particles were processed further using non-uniform refinement to obtain the final cryo-EM maps for each class.

To improve the quality of the cryo-EM maps around the ICDs, we performed local refinements using masks that cover the ICD of each subunit (*Figure 2—figure supplement 2*). We observed a noticeable improvement in class 1 structure and used these maps for model building. We applied the same strategy for the LRRDs. However, the quality of the maps did not improve to a level allowing us to build reliable models.

Data processing for 8C-8A(IL1$^{25}$) nanodiscs was performed as described above, with the following exceptions. Motion correction was performed using MotionCor2 (*Zheng et al., 2017*) in Relion 3.0 (*Zivanov et al., 2018*). The final set of particles was extracted with a box size of 480×480 pixels, and they were classified into three 3D classes.

## Model building

Models were built using Coot (*Emsley and Cowtan, 2004*). We initially placed the human LRRC8C model from the AlphaFold protein structure prediction database (*Jumper et al., 2021*; *Varadi et al., 2022*) in the class 1 density. We manually fitted the individual residues into the density while removing the parts that did not have corresponding interpretable densities. Once we built a complete subunit, we copied the model into the other six protomers and manually fit individual residues into the density. The model was refined using Phenix real-space refinement (*Afonine et al., 2018*). We performed iterative build-refine cycles till a satisfactory model was obtained. The resulting model was fitted into the class 2 and 3 maps and fitted into the density using the same build-refine iterations as described above. Because of their limited resolution, most parts of class 5 and 6 structures were modeled without their side chains (i.e., as alanines) while maintaining their correct labeling for the amino acid type. The LRRDs were not modeled in all five structures. 8A-IL1$^{25}$ is longer by four amino acids than the corresponding region in LRRC8C (*Figure 1*). Therefore, the residue numbering after 8A(IL1$^{25}$) in the 8C-8A(IL1$^{25}$) chimera increases by four amino acids in the polypeptide. However, we kept the LRRC8C numberings in the models to facilitate easier comparison with other studies. Validations of the structural models were performed using MolProbity (*Williams et al., 2018*) implemented in Phenix (*Afonine et al., 2018*).

Some of the data processing and refinement software was supported by SBGrid (*Morin et al., 2013*).

## Patch-clamp electrophysiology

LRRC8A and 8C-8A(IL1$^{25}$) cDNA constructs were expressed in *Lrrc8*$^{-/-}$ HCT116 cells transfected using Turbofectin 8.0 (OriGene Technologies) with 0.125 μg GFP cDNA and 0.25 μg of 8A and 8C-8A(IL1$^{25}$).

Transfected cells were identified by GFP fluorescence and patch clamped in the whole-cell mode at room temperature using patch electrodes pulled from 1.5 mm outer-diameter silanized borosilicate

**Table 3.** Composition of patch pipette and solutions.

| | Patch pipette solutions | | Bath solutions | |
|---|---|---|---|---|
| | Control | Control | Control | Hypotonic |
| CsCl | 126 mM | 26 mM | 75 mM | 75 mM |
| Cesium methanesulfonate | | 100 mM | | |
| $MgSO_4$ | 2 mM | 2 mM | 5 mM | 5 mM |
| Ca-gluconate$_2$ | | | 1 mM | 1 mM |
| ATP-Na$_2$ | 2 mM | 2 mM | | |
| GTP-Na$_2$ | 0.5 mM | 0.5 mM | | |
| Glutamine | | | 2 mM | 2 mM |
| EGTA | 1 mM | 1 mM | | |
| HEPES | 20 mM | 20 mM | 12 mM | 12 mM |
| Tris | | | 8 mM | 8 mM |
| CsOH | 12 mM | 12 mM | | |
| HCl | | | 2 mM | 2 mM |
| Glucose | | | 5 mM | 5 mM |
| Sucrose | 16 mM | 16 mM | 115 mM | 70 mM |
| pH* | 7.2 | 7.2 | 7.4 | 7.4 |
| Osmolality | 275 mOsm | 275 mOsm | 300 mOsm | 250 mOsm |
| Ionic strength | 0.162 M | 0.162 M | | |

*The pH of patch pipette and bath solutions was adjusted with CsOH and HCl, respectively.

microhematocrit tubes. Recordings were not performed on cells where access resistance was >2-fold that of the pipette resistance.

The composition of the control bath and pipette solutions used in these studies is shown in *Table 3*. Intracellular ionic strength was reduced by the removal of CsCl or cesium methanesulfonate from the patch pipette solution and replacement with sucrose to maintain solution osmolality.

Whole-cell currents were measured with an Axopatch 200A (Axon Instruments) patch-clamp amplifier. Electrical connections to the patch-clamp amplifier were made using Ag/AgCl wires and 3 M KCl/ agar bridges. Series resistance was compensated by >85% to minimize voltage errors. Data acquisition and analysis were performed using pClamp 10 software (Axon Instruments).

Changes in current amplitude were quantified using a voltage-ramping protocol. Membrane voltage was held at –30 mV throughout all experiments. Ramps were initiated by stepping membrane voltage to –100 mV and then ramping membrane voltage over 1 s to +100 mV. This was followed by a step back to –30 mV for 4 s before the ramp was repeated.

Relative anion permeability ($P_X/P_{Cl}$) was measured from Cl$^-$ substitute-induced changes in reversal potential using a modified Goldman-Hodgkin-Katz equation (*Voss et al., 2014*). LRRC8A and 8C-8A(IL1[25]) currents were activated with low ionic strength, 0.062 M, patch pipette solution, and by cell swelling induced by reducing bath osmolality to 250 mOsm. After stable current activation was achieved, changes in reversal potential were induced by replacing bath CsCl with either cesium glutathione or cesium lactobionate. Reversal potentials were corrected for anion-induced changes in liquid junction potentials.

Electrophysiological data are presented as means ± SEM. All patch-clamps studies were performed on at least two independently transfected groups of cells. n represents the number of patch-clamped cells from which currents were recorded. Statistical significance was determined using Student's $t$ test for unpaired means.

## Figure preparation

Figures were prepared using Chimera (*Pettersen et al., 2004*), ChimeraX (*Pettersen et al., 2021*), and The PyMOL Molecular Graphics System (Version 2.0, Schrödinger, LLC). Calculation of the pore radii was performed using the software HOLE (*Smart et al., 1996*).

## Acknowledgements

We thank Dr Kunpeng Lee at Case Western Reserve University and Melissa Chambers and Scott Collier at the Cryo-EM facility at Vanderbilt University for cryo-EM data collection. This work was conducted in part using the CPU and GPU resources of the Advanced Computing Center for Research and Education (ACCRE) at Vanderbilt University. We used the DORS storage system supported by the National Institute of Health (NIH) (S10 RR031634 to Jarrod Smith). This work was supported by the National Institute of Diabetes, Digestive, and Kidney Diseases Grant R01 DK51610 to JSD.

## Additional information

### Competing interests

Kevin Strange: is cofounder and principal scientist of Revidia Therapeutics, Inc. The other authors declare that no competing interests exist.

### Funding

| Funder | Grant reference number | Author |
|---|---|---|
| National Institute of Diabetes and Digestive and Kidney Diseases | DK51610 | Jerod S Denton |

The funders had no role in study design, data collection and interpretation, or the decision to submit the work for publication.

### Author contributions

Hirohide Takahashi, Toshiki Yamada, Data curation, Formal analysis, Investigation, Methodology, Writing – review and editing; Jerod S Denton, Conceptualization, Formal analysis, Supervision, Funding acquisition, Validation, Project administration, Writing – review and editing; Kevin Strange, Conceptualization, Formal analysis, Supervision, Funding acquisition, Writing – original draft, Project administration, Writing – review and editing; Erkan Karakas, Conceptualization, Formal analysis, Supervision, Funding acquisition, Validation, Visualization, Writing – original draft, Project administration, Writing – review and editing

### Author ORCIDs

Hirohide Takahashi http://orcid.org/0000-0002-2553-8806
Erkan Karakas http://orcid.org/0000-0001-6552-3185

### Decision letter and Author response

Decision letter https://doi.org/10.7554/eLife.82431.sa1
Author response https://doi.org/10.7554/eLife.82431.sa2

## Additional files

### Supplementary files
• MDAR checklist

## Data availability

Cryo-EM maps and atomic coordinates are deposited to the Electron Microscopy Data Bank (EMDB) and Protein Data Bank (PDB) databases. The accession codes are EMD-27770 and 8DXN for class 1; EMD-27771 and 8DXO for class 2; EMD-27772 and 8DXP for class 3; EMD-27773 and 8DXQ for class 4; EMD-27774 and 8DXR for class 5.

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
