## [Editor Report]

This paper is valuable to the field of ion channels, as it provides useful information about the assembly of the volume-regulated anions channels formed by LRRC8 proteins. The evidence that the LRRC8C-LRRC8A chimera has native functional properties and adopts a heptameric assembly is convincing. The evidence supporting the physiological relevance of the heptameric assembly and the hypothesized role of lipids is incomplete and will be addressed in future studies.

---

## [Decision Letter]

**Decision letter after peer review:**

Thank you for submitting your article "A LRRC8 chimera with native functional properties is a heptamer with a large lipid-blocked pore" for consideration by *eLife*. Your article has been reviewed by 3 peer reviewers, and the evaluation has been overseen by a Reviewing Editor and Richard Aldrich as the Senior Editor. The reviewers have opted to remain anonymous.

Overall, the reviewers agree that the structural and functional characterization of the chimera is interesting from a channel architecture and assembly point of view but that the physiological relevance of the heptamer and the proposed lipid gating are not sufficiently supported by the data. As you will see below, the reviewers have recommended potential experiments to strengthen your conclusions. However, the consensus is that fully addressing these points would require extensive additional work, and therefore our recommendation is that a revised version of this manuscript should focus on the characterization of the chimera and not extend conclusions to physiological channel assemblies.

Essential revisions:

1) Physiological relevance:

Given that other LRRC8 channels are hexamers, the relevance of the heptameric assembly of the chimeric construct analyzed here should be experimentally evaluated. For example, the authors should identify structural elements in the heptameric assembly that account for physiological properties of native channels. Are there any functional channel properties that are explained by the heptameric assembly that are not compatible with the hexameric one (or viceversa?). The only evidence presented thus far is the increase in the permeability ratios of gluthatione and lactobionate between LLRC8A and 8C-8A(IL125). However, these effects are quite small, <50% increase between the two channels. Further, the LLRC8A homomeric pore is permeable to these ions. This seems at odds with its reported ~2 A radius, suggesting that the open homomeric channel is wider than the reported close conformation. Additional pharmacological and/or mutagenesis data is needed to corroborate this.

Native LRRC8 channels consist of the obligatory LRRC8A subunit and other paralogs (LRRC8B-E). The chimera in this manuscript is constructed on the basis of LRRC8C, with a swapped intracellular loop from LRRC8A that consists of 25 amino acids. It is interesting that the chimera recapitulates some native channel properties and forms a distinct heptameric channel, but this chimera is drastically different from native channels because the lack of an intact LRRC8A subunit. For instance, the selectivity filter arginine R103 in LRRC8A corresponds to L105 in LRRC8C. Thus, the pore properties of the chimera with all Leucine residues at this position would be very different from native LRRC8 channels. Therefore, the physiological relevance is unclear.

The authors propose that native heteromeric LRRC8 channels form heptamers based on the heptameric structure of the chimera. However, previous studies (Deneka et al. 2018, Nature) have shown that the functional heteromeric LRRC8A/C channels also form hexamers. In addition, two recent preprints corroborated the hexameric assembly of LRRC8A/C heteromeric channels.

2) Lipid characterization and function:

As shown in Figure 6b, the cryoEM density appears to be shown at very low threshold (as assessed by the micelle density), the assigned lipid-like density is similar to the micelle density. We are concerned that these are just noisy density that does not represent ordered/occupied lipids in the channel.

Further, the evidence supporting pore occlusion by lipids is weak. Based on the hydrophobic profile of the pore, it is difficult to visualize how the lipids can rearrange into the pore in two layers. Even if lipids are blocking the pore, authors cannot rule out that this is an artifact because of the purification process. Although they performed channel reconstitution in nanodiscs, the lipid-densities inside the pore can still be artifactual due to significant amounts of detergents that are used during the purification process. In addition, even when lipids-like densities have been found inside the pore of other channels, there is not functional evidence for supporting a gating role in large-pore channels and current hypotheses are still speculative. Thus, it is hard to visualize a gating mechanism by lipids in VRAC channels; if pore occlusion by lipids occurs in native cells, how the lipids can move in and out from the pore during a gating event? Overall, without proper data, it is not suitable to conclude that lipids are essential for normal channel gating and regulation of VRAC.

Experiments aimed at supporting the idea that this assembly is physiologically relevant are essential for supporting the authors' conclusion that "associated lipids are essential for normal channel gating and regulation." For example, the authors could show that the 8C-8A(IL125) channels co-purify with many more lipids than the homomeric 8A ones using lipidomics and/or native mass spec. Alternatively, functional experiments showing that lipids affect not only channel gating, but also ion permeation and selectivity would greatly strengthen the authors' conclusion.

3) Pore properties:

The authors argued that chimera structures reveal that the narrower part of pore is wide enough (~ 5 Å radius) for the permeation of molecules like glutathione and lactobionate. Further evidence is needed to support this conclusion, such as molecular modeling. The functional data alone are not sufficient to support that the new structure corresponds to the molecule-permeable conformation. In addition, the electrophysiological data are poorly described/presented. What is the reversal potential of non-transfected cells under the same experimental conditions?

A very interesting finding of this work is the heterogeneity observed in different protomers; specifically, a switch between a positive charged residue (K51) and a negative charged residue (D50) at the entrance of the pore. Considering that these residues are conserved in the LRRC8A and LRRC8E, it would be very interesting to evaluate how these residues affect gating and permeability properties of the chimera and LRRC8A channels.

An intriguing question that arises after reading this manuscript is related the mechanism blockade of DCPIB in VRAC channels. Previous structural work proposed that this blocker occludes the pore by interacting with residue R103, but this residue is only present in LRRC8A and B channels. LRRC8 C, D and E do not have an arginine at this position. In a previous manuscript by coauthors (Yamada et al., Am J Physiol Cell Physiol. 2021), the apparent affinity for the DCPIB is higher in the LRRC8 chimeric suggesting a different blockade mechanism to that previously proposed. How could the authors explain these results?

Presentation Clarity:

1) Pg. 4 The authors should tone down some of their claims. For example, they claim "…five high resolution (3.3-4.0 A) structures…" Given current standards, I do not think that 3.3-4 Å is not 'high resolution'. Similarly, they state "…allowing us to build an atomic model comprising most of the protein, except residue 60-94 in the ECD and the first 15 residues of the N-terminus." This is not a proper description of the data as the very large LLRD domains are not resolved in their structures, so that most of the protein could not actually be resolved.

2) Please find a different nomenclature than 'wide' and 'narrow' interface. I find these adjectives to be confusing and somewhat misleading descriptors of the structural features of LRRC8 channels. A wide/narrow interface suggest that the interface is wide or narrow. Instead, it is the separation between the protomers that is wide or narrow. Thus, the interface surface in the 'wide' arrangement is smaller than in the 'narrow' conformation, further confusing a reader. Maybe 'tight' and 'loose' would be better descriptors of the interfaces observed?

3) Please expand on the differences in the TMD regions of the different classes. When the authors say "the clustered arrangement is preserved" do the authors mean the 4 and 3 subunit arrangement? if yes, this is not apparent is the data shown in Supp Figure 5.

4) What are the consequences of the relative changes in protomer positions in the 5 classes? How do the interfaces change? An analysis of interaction surfaces in the 5 classes would be helpful to clarify what the differences are.

5) Are the lipid-like densities between protomers the same in the narrow and wide interfaces? Are the densities the same in all 7 protomers and 5 classes, despite the differences in interaction surfaces? This should be documented and discussed.

6) All structural figures show the intracellular domains at the top and the extracellular domains at the bottom. This is opposite to the conventional orientation of the plasma membrane of a cell.

7) Based on their current data, the authors suggest that formation of native heteromeric VRAC channels are heptamers. It is possible, however, that different LRRC8 isoforms can have different oligomeric composition (hexamer and heptamer) and heptamers can be found for LRRC8C homomeric channels. An example of how different isoforms from the same family can different oligomer are the CALHM channels. This should be discussed.

8) The manuscript lacks proper data description/analysis of the structural basis for an heptameric assembly. For example, what are the interactions between subunits in the chimera? What key differences are observed with the already known LRRC8 hexameric structures?

9) Bands in the native page blot shown in Figure Sup. 1, are not clear (especially the chimera). Do the chimeric channels form exclusively heptamers or also can form another oligomeric state?

10) Figure Supplementary 2 and 7. FSC plot quality is quite low and almost unreadable. In addition, the scale of Y-axis in Figure Sup. 2 is apparently wrong.

[Editors’ note: further revisions were suggested prior to acceptance, as described below.]

Thank you for resubmitting your work entitled "A LRRC8 chimera with native functional properties is a heptamer with a large-diameter lipid-blocked pore" for further consideration by *eLife*. Your revised article has been evaluated by Richard Aldrich (Senior Editor) and a Reviewing Editor.

This new version of the manuscript has considerably toned down the conclusions from the resolved structures and incorporated new data, making it more focused on the description/characterization of this novel structure. There are only a few remaining issues that need to be addressed, as outlined below:

2) The evidence presented does not definitively demonstrate that the pore is completely occluded by lipid, as suggested in the title of this manuscript and the Result section (page 9, lines 209-211). We request that the title be changed and the Results section reworded in order to avoid potential confusion in the field.

3) Additional thoughts on the point-by-point letter concerning lipid densities:

Author statement: "To clarify the representation of the lipid-like densities, we modified Figure 6 (Figure 7 in the revised manuscript) as follows. The threshold of the cryo-EM map was increased. We aligned the panels showing the cryo-EM density, hydrophobicity plot, and electrostatic surface charges side by side to reveal how the lipid-like density aligns with the chemical properties of the inner surface of the pore. We used PyMol to prepare hydrophobicity and electrostatic surface potential figures in the revised manuscript."

From the figures, it is still unclear how the lipid-like densities align with the hydrophobicity/electrostatic profile of the pore. Surface pore properties are not correlative with the formation of two lipid layers. Also, a better representation of lipid-like densities between subunits is needed. A zoomed image could help to show this better. If authors can identify a conserved hydrophobic pocket for these lipid interactions, that could be highlighted. This point is not critical, but the manuscript may benefit by providing further information on the residues interacting with lipid-like densities between subunits.

4) You have included new figures highlighting the tight and loose interfaces in ECD, TMD, and ICD (Figure 3 —figure supplement 1-3). This is interesting, but an accurate description of the main interactions is not found in the main text.

5) You could elaborate on the description of D50/K51 rearrangements in the Result section. You state, "When K51 points toward the pore, it creates a groove between the subunits (Figure 4D)". Although this statement seems correct in the current context, the sentence confuses the overall interpretation of D50/K51 rearrangements in LRRC8C channels because the statement only applies for structures of class 1 and 3 but not for classes 2, 4, and 5. For example, the structure of class 2 only has one K51 side chain pointing toward the pore (Figure 5 – Figure sup2). Still, it keeps two loose interfaces anyway (Figure 5C), which suggests that D50/K51 rearrangement is not necessarily coupled with the loose interface as one could interpret from the text.

6) Page 4, Line 80-82 "The oligomeric state directly impacts pore size, pharmacology, and permeability to large anionic solutes. Our results also suggest that associated membrane lipids may be involved in channel gating and regulation".

This work did not evaluate if the oligomeric state affects the pharmacology of the channels. As discussed in the point-by-point letter, current evidence supports the hypothesis that pore size could impact pharmacology, but this could also be related to the intrinsic properties of LRRC8C. Also, in view that there is no evidence supporting the role of lipids in channel function, the above sentence is speculative. Thus, this entire sentence should be restricted to the Discussion section only.

7) Figure 1: Please Indicate in the figure legend what the symbols shown below the alignment in panel A (*. :) mean.

8) Please specify in the figure legends what class is shown in figures 2, 3, and 4. Are these the class 1 structure?

9) Figure 4: The arrow showing rotation between structures of panels D and E seems to be in the wrong direction. Please check.

10) Figure legend 7: "B) A sliced view of the unsharpened cryo-EM map (grey, transparent) with the ribbon representation of the 8C-8A(IL125) structure".

The ribbon representation is not shown in the new version of the manuscript. Please, check.

11) Table 3: The concentration unit for sucrose concentration is missing. Please check.

---

## [Author Response]

Essential revisions:1) Physiological relevance:Given that other LRRC8 channels are hexamers, the relevance of the heptameric assembly of the chimeric construct analyzed here should be experimentally evaluated. For example, the authors should identify structural elements in the heptameric assembly that account for physiological properties of native channels. Are there any functional channel properties that are explained by the heptameric assembly that are not compatible with the hexameric one (or viceversa?). The only evidence presented thus far is the increase in the permeability ratios of gluthatione and lactobionate between LLRC8A and 8C-8A(IL125). However, these effects are quite small, <50% increase between the two channels. Further, the LLRC8A homomeric pore is permeable to these ions. This seems at odds with its reported ~2 A radius, suggesting that the open homomeric channel is wider than the reported close conformation. Additional pharmacological and/or mutagenesis data is needed to corroborate this.Native LRRC8 channels consist of the obligatory LRRC8A subunit and other paralogs (LRRC8B-E). The chimera in this manuscript is constructed on the basis of LRRC8C, with a swapped intracellular loop from LRRC8A that consists of 25 amino acids. It is interesting that the chimera recapitulates some native channel properties and forms a distinct heptameric channel, but this chimera is drastically different from native channels because the lack of an intact LRRC8A subunit. For instance, the selectivity filter arginine R103 in LRRC8A corresponds to L105 in LRRC8C. Thus, the pore properties of the chimera with all Leucine residues at this position would be very different from native LRRC8 channels. Therefore, the physiological relevance is unclear.The authors propose that native heteromeric LRRC8 channels form heptamers based on the heptameric structure of the chimera. However, previous studies (Deneka et al. 2018, Nature) have shown that the functional heteromeric LRRC8A/E channels also form hexamers. In addition, two recent preprints corroborated the hexameric assembly of LRRC8A/C heteromeric channels.

It is unclear at present whether native LRRC8 channels are hexamers, heptamers or whether they exist in multiple conformations including higher order assemblies. We have reduced the discussion of the physiological relevance of the heptameric assembly and instead point out the need for much additional study to understand the configuration of native channels.

We agree that additional pharmacological and mutagenesis data are needed to address this important problem. Some of these data will be presented at the upcoming Biophysical Society meeting and will be the focus of a forthcoming manuscript. However, as we discussed in the current manuscript, we have shown previously that the DCPIB pharmacology of hexameric LRRC8A channels is grossly abnormal compared to native channels and 8A+8C heteromers. DCPIB pharmacology of 8A hexamers is also dramatically altered by mutation of R103 to phenylalanine. R103 forms the narrow constriction of 8A hexameric channels. In contrast, 8A+8C heteromers are unaffected by the R103F mutation.

The two new manuscripts that have appeared on bioRxiv show clear hexameric structures of 8A+8C heteromers. However, the 8A:8C stoichiometries differ strikingly for the two labs. The Dutzler lab shows a 4A:2C stoichiometry, while the Brohawn lab shows a 5A:1C stoichiometry. It is important to note that four labs have shown that 8A functions as a dominant-negative. High levels of 8A expression completely block native LRRC8 currents and dramatically reduce heterologously expressed channel activity. Thus, one can well imagine that high levels of 8A expression drive the formation of non-native channel configurations. This is an important problem that must be addressed in order to understand the configurations and stoichiometries of native channels.

Finally, it is important to note that the Dutzler lab states in their bioRxiv manuscript that it is unclear how a hexameric configuration can give rise to a large pore channel permeable to diverse organic solutes.

2) Lipid characterization and function:As shown in Figure 6b, the cryoEM density appears to be shown at very low threshold (as assessed by the micelle density), the assigned lipid-like density is similar to the micelle density. We are concerned that these are just noisy density that does not represent ordered/occupied lipids in the channel.Further, the evidence supporting pore occlusion by lipids is weak. Based on the hydrophobic profile of the pore, it is difficult to visualize how the lipids can rearrange into the pore in two layers. Even if lipids are blocking the pore, authors cannot rule out that this is an artifact because of the purification process. Although they performed channel reconstitution in nanodiscs, the lipid-densities inside the pore can still be artifactual due to significant amounts of detergents that are used during the purification process. In addition, even when lipids-like densities have been found inside the pore of other channels, there is not functional evidence for supporting a gating role in large-pore channels and current hypotheses are still speculative. Thus, it is hard to visualize a gating mechanism by lipids in VRAC channels; if pore occlusion by lipids occurs in native cells, how the lipids can move in and out from the pore during a gating event? Overall, without proper data, it is not suitable to conclude that lipids are essential for normal channel gating and regulation of VRAC.Experiments aimed at supporting the idea that this assembly is physiologically relevant are essential for supporting the authors' conclusion that "associated lipids are essential for normal channel gating and regulation." For example, the authors could show that the 8C-8A(IL125) channels co-purify with many more lipids than the homomeric 8A ones using lipidomics and/or native mass spec. Alternatively, functional experiments showing that lipids affect not only channel gating, but also ion permeation and selectivity would greatly strengthen the authors' conclusion.

We agree with the reviewers that additional experiments are required to conclude that these densities are lipid molecules and that lipids are essential for normal channel gating. Therefore, we have modified the text and presented our observation as a possible mechanism that requires further experimental validation.

As pointed out, the cryo-EM density we assign as lipid-like density is not as strong as the protein densities. However, we think this would be expected since most lipid molecules within the pore would not interact with the protein residues and be highly flexible, resulting in weaker cryo-EM densities similar to the ones for the detergent micelle. Moreover, the "lipid-like densities" in the nanodisc structures are highly similar, both in intensity and shape, to the lipid densities observed for the lipid bilayer around the protein within the nanodisc, as shown in Figure 7—figure supplement 1 (Supp. Figure 7 in the original submission). In agreement with our observations, in a recent preprint reporting the hexameric structure of LRRC8A-LRRC8C (Kern et al. 2022), ordered lipid molecules are observed on the extracellular side of the pore opening, aligning well with the densities we observed. It is plausible that the narrower opening in the hexameric structure compared to that of the heptameric structure led to the visualization of the ordered lipid molecules.

To clarify the representation of the lipid-like densities, we modified Figure 6 (Figure 7 in the revised manuscript) as follows. The threshold of the cryo-EM map was increased. We aligned the panels showing the cryo-EM density, hydrophobicity plot, and electrostatic surface charges side by side to reveal how the lipid-like density aligns with the chemical properties of the inner surface of the pore. We used PyMol to prepare hydrophobicity and electrostatic surface potential figures in the revised manuscript.

3) Pore properties:The authors argued that chimera structures reveal that the narrower part of pore is wide enough (~ 5 Å radius) for the permeation of molecules like glutathione and lactobionate. Further evidence is needed to support this conclusion, such as molecular modeling. The functional data alone are not sufficient to support that the new structure corresponds to the molecule-permeable conformation. In addition, the electrophysiological data are poorly described/presented. What is the reversal potential of non-transfected cells under the same experimental conditions?

Our intent was to show that there are clear differences in the relative permeabilities of these two solutes in the 8C-8A(IL1^25^) heptameric channel when compared directly to 8A hexameric channels. Our results demonstrate that the relative permeability to these solutes is significantly higher in the chimera. This is consistent with the larger pore diameter of the heptameric chimera versus the 8A hexamer. This section has now been rewritten to improve clarity.

Countless labs have measured relative permeabilites of native LRRC8/VRAC channels to a host of diverse solutes. The estimated pore diameter of native channels is similar to that of 8C-8A(IL1^25^) heptameric channels. In contrast, the narrow pore diameters of 8A hexamer as well as the 8A+8C heteromers described in the recent manuscripts from the Brohawn and Dutzler labs are well below that estimated for native channels. As noted above, the Dutzler lab states in their bioRxiv manuscript that it is unclear how a hexameric configuration can give rise to a large pore channel permeable to diverse organic solutes.

A very interesting finding of this work is the heterogeneity observed in different protomers; specifically, a switch between a positive charged residue (K51) and a negative charged residue (D50) at the entrance of the pore. Considering that these residues are conserved in the LRRC8A and LRRC8E, it would be very interesting to evaluate how these residues affect gating and permeability properties of the chimera and LRRC8A channels.

To evaluate the effect of the switch between K51 and D50 at the entrance of the pore, we modified Figure 4 (Figure 3 in the original manuscript) by adding 3 new panels. These new panels show the surface representation of the protein, highlighting the differences in the spacing between the subunits due to this switch. We also modified the text accordingly.

An intriguing question that arises after reading this manuscript is related the mechanism blockade of DCPIB in VRAC channels. Previous structural work proposed that this blocker occludes the pore by interacting with residue R103, but this residue is only present in LRRC8A and B channels. LRRC8 C, D and E do not have an arginine at this position. In a previous manuscript by coauthors (Yamada et al., Am J Physiol Cell Physiol. 2021), the apparent affinity for the DCPIB is higher in the LRRC8 chimeric suggesting a different blockade mechanism to that previously proposed. How could the authors explain these results?

We believe that the larger pore diameter of native LRRC8 channels, heterologously expressed 8A+8C heteromers, and the 8C-8A(IL1^25^) heptameric chimera is the simplest explanation for the difference. Notably, the Hill coefficient for DCPIB is close to 1 for narrow pore 8A hexameric channels suggesting that a single DCPIB molecule blocks the pore. This would be consistent with the cryo-EM structure published by Kern et al. showing that the pore is occluded by a single DCPIB molecule at its narrowest constriction formed by R103. In contrast, the Hill coefficients for the chimera and native channels are >2, suggesting multiple cooperative DCPIB binding sites.

Also of note is the effect of R103 mutations on DCPIB inhibition. Mutation of R103 to phenylalanine dramatically reduces DCPIB inhibition of narrow pore 8A hexamers. This is predicted due to the further narrowing of the already narrow pore by the phenylalanine. In contrast, R103F has no effect on DCPIB inhibition of 8A(R103F)+8C heteromers (Yamada et al.). Phenylalanine mutation of the equivalent residue, L105, has no effect on DCPIB inhibition of the 8C-8A(IL1^25^) heptameric chimera or on 8A(R103F)+8C(L105F) heteromers. These latter data have been published in abstract form and are the subject of a forthcoming manuscript.

Finally, DCPIB inhibition is strongly voltage-dependent in 8A hexameric channels, whereas it is voltage insensitive in native channels, 8A+8C heteromers, and 8C-8A(IL1^25^) heptameric chimeras. The voltage sensitivity of narrow pore 8A hexamers is most readily explained by the DCPIB block at the channel constriction site formed by R103.

Presentation Clarity:1) Pg. 4 The authors should tone down some of their claims. For example, they claim "…five high resolution (3.3-4.0 A) structures…" Given current standards, I do not think that 3.3-4 Å is not 'high resolution'. Similarly, they state "…allowing us to build an atomic model comprising most of the protein, except residue 60-94 in the ECD and the first 15 residues of the N-terminus." This is not a proper description of the data as the very large LLRD domains are not resolved in their structures, so that most of the protein could not actually be resolved.

We have changed the text to tone down the claims as follows:

"….Therefore, the final reconstructions were done without enforcing any symmetry and resulted in five structures with resolutions in the range of 3.3 to 4.0 Å (Figures 1a,b; Supplementary Figure 2-3)."

"…The cryo-EM maps for the ECD and TMD revealed high resolution features allowing us to build an atomic model comprising most of the ECD and TMD except residues 60-94 in the ECD and the first 15 residues of the N-terminus…."

2) Please find a different nomenclature than 'wide' and 'narrow' interface. I find these adjectives to be confusing and somewhat misleading descriptors of the structural features of LRRC8 channels. A wide/narrow interface suggest that the interface is wide or narrow. Instead, it is the separation between the protomers that is wide or narrow. Thus, the interface surface in the 'wide' arrangement is smaller than in the 'narrow' conformation, further confusing a reader. Maybe 'tight' and 'loose' would be better descriptors of the interfaces observed?

We have changed the nomenclature of "wide" and "narrow" interfaces to "loose" and "tight" interfaces, respectively, on the text and the figures.

3) Please expand on the differences in the TMD regions of the different classes. When the authors say "the clustered arrangement is preserved" do the authors mean the 4 and 3 subunit arrangement? if yes, this is not apparent is the data shown in Supp Figure 5.

We added a panel to Figure 5 (Figure 4 in the original manuscript) to compare the TMD structures in all five classes. We used reference points to compare the separation between the neighboring subunits, along with the comparison of the ICD structures. We also added a new panel to Figure 5-supplement 1 (Supp. Figure 5 in the original manuscript), showing the alignment of all five structures at the TMD to compare the structures. We removed the phrase "the clustered arrangement is preserved" and emphasized the similarity to the differences in subunit separation at the loose and tight interfaces to that of ICD.

4) What are the consequences of the relative changes in protomer positions in the 5 classes? How do the interfaces change? An analysis of interaction surfaces in the 5 classes would be helpful to clarify what the differences are.

The main effect of the changes in the protomer positions is the separation between the subunits at the TMD and ICD. The changes are more robust at the loose interfaces and likely affect how the interior and outside of the pore are connected through the gaps between the subunits. The new Figure 5 (Figure 4 in the original manuscript) shows the changes in subunit separation at the ICD and TMD in the five classes.

5) Are the lipid-like densities between protomers the same in the narrow and wide interfaces? Are the densities the same in all 7 protomers and 5 classes, despite the differences in interaction surfaces? This should be documented and discussed.

The densities at the loose cytoplasmic side of the loose interfaces are considerably weaker than the ones at the tight interfaces. We changed Figure 7A (Figure 6A in the original manuscript) to show this difference and edited the text accordingly.

6) All structural figures show the intracellular domains at the top and the extracellular domains at the bottom. This is opposite to the conventional orientation of the plasma membrane of a cell.

All the figures are changed to place the extracellular domains at the top and the intracellular domains at the bottom.

7) Based on their current data, the authors suggest that formation of native heteromeric VRAC channels are heptamers. It is possible, however, that different LRRC8 isoforms can have different oligomeric composition (hexamer and heptamer) and heptamers can be found for LRRC8C homomeric channels. An example of how different isoforms from the same family can different oligomer are the CALHM channels. This should be discussed.

We included the following paragraphs in the manuscript to discuss the possibility of multiple oligomeric states for LRRC8 channels.

8) The manuscript lacks proper data description/analysis of the structural basis for an heptameric assembly. For example, what are the interactions between subunits in the chimera? What key differences are observed with the already known LRRC8 hexameric structures?

To expand on the description of the subunit interfaces, we included 3 additional supplementary figures (Figure 3-supplement 1-3) showing detailed view of the interfaces between the subunits at the loose and tight interfaces of the ECD, TMD, and ICD. We also calculated the buried surface areas between the subunits for each domain in the class 1 structure.

“Two recent studies have defined high-resolution cryo-EM structures of LRRC8A/LRRC8C heteromeric channels (Kern et al., 2022; Rutz et al., 2022). Both studies demonstrated that the heteromeric channels could adopt a hexameric conformation with a limiting pore radius of 2-3 Å, similar to that of LRRC8A hexamers. However, these channels either had 5:1 (Kern et al., 2022) or 4:2 (Rutz et al., 2022) LRRC8A:LRRC8C stoichiometries, indicating that LRRC8A:LRRC8C heteromers can adopt multiple oligomeric forms.

Taken together, existing cryo-EM structural data suggest that native VRAC/LRRC8 channels may exist in multiple oligomeric conformations, as has been reported for CALHM channels. It will be critical to relate the functional properties of various channel oligomeric conformations to those of native VRAC/LRRC8 channels. It will also be important to define the role of LRRC8A in channel assembly and conformation. Four groups have shown that LRRC8A has a dominant-negative function. Overexpression of LRRC8A suppresses endogenous (Qiu et al., 2014; Syeda et al., 2016; Voss et al., 2014) and heterologously expressed (Yamada et al., 2016) VRAC/LRRC8 currents. LRRC8A protein levels may therefore impact channel assembly and function.”

9) Bands in the native page blot shown in Figure Sup. 1, are not clear (especially the chimera). Do the chimeric channels form exclusively heptamers or also can form another oligomeric state?

The Flag tag on the constructs appear to cause smeary bands on the native PAGE blots, but 8A and the chimera run separately in all experiments we have done. We also do not observe any classes that suggested other possible stoichometries in our cryo-EM experiments and obtain single peaks in the size exclusion chromatography experiments suggesting that the chimeric channels are mainly heptameric.

10) Figure Supplementary 2 and 7. FSC plot quality is quite low and almost unreadable. In addition, the scale of Y-axis in Figure Sup. 2 is apparently wrong.

We enlarged the FSC plots so that they are more easily readable. We also corrected the labeling of the Y-axis in Supplemental Figure 2.

[Editors’ note: further revisions were suggested prior to acceptance, as described below.]

2) The evidence presented does not definitively demonstrate that the pore is completely occluded by lipid, as suggested in the title of this manuscript and the Result section (page 9, lines 209-211). We request that the title be changed and the Results section reworded in order to avoid potential confusion in the field.

We changed the title to "Cryo-EM structures of a LRRC8 chimera with native functional properties reveal heptameric assembly" and reworded the Results section. It should now be clear to the reader that further experiments are required to verify the presence of pore-blocking lipid molecules.

3) Additional thoughts on the point-by-point letter concerning lipid densities:Author statement: "To clarify the representation of the lipid-like densities, we modified Figure 6 (Figure 7 in the revised manuscript) as follows. The threshold of the cryo-EM map was increased. We aligned the panels showing the cryo-EM density, hydrophobicity plot, and electrostatic surface charges side by side to reveal how the lipid-like density aligns with the chemical properties of the inner surface of the pore. We used PyMol to prepare hydrophobicity and electrostatic surface potential figures in the revised manuscript."From the figures, it is still unclear how the lipid-like densities align with the hydrophobicity/electrostatic profile of the pore. Surface pore properties are not correlative with the formation of two lipid layers. Also, a better representation of lipid-like densities between subunits is needed. A zoomed image could help to show this better. If authors can identify a conserved hydrophobic pocket for these lipid interactions, that could be highlighted. This point is not critical, but the manuscript may benefit by providing further information on the residues interacting with lipid-like densities between subunits.

We have made several changes to Figure 7 and the text titled "Interaction of lipids with the 8C-8A(IL1^25^)" on page 9. Specifically;

– We changed the color of lipid-like densities between the subunits to yellow from red to avoid potential confusion with the red color used to show the negative charge.

– We added a new panel (panel B) showing the zoomed view of one of the lipid-like densities between the subunits. In this figure, the cryo-EM map was shown as a grey mesh. The density was not trimmed to a particular region in this figure so that the readers can judge the quality of the lipid density relative to the surrounding region. Although we did not model any ligands to the deposited structures, we placed a phospholipid into the density for illustrative purposes. Select residues in proximity to the phospholipid molecule were shown.

– In the panel showing the surface properties of the pore lining residues (panels C and D in the revised manuscript), we included hypothetical lipid molecules that would form the bilayer around the TMD to emphasize how surface properties align with the membrane. The positioning of the membrane boundaries around the protein is calculated using PPM 2.0 (https://opm.phar.umich.edu/ppm_server) incorporated into CHARMM-GUI (https://www.charmm-gui.org/?doc=input/membrane.bilayer).

– We changed the coloring of the hydrophobicity from red to orange to avoid potential confusion with the red color used to show electronegative surface charge.

– We added a new panel showing a hypothetical representation of the pore with lipid molecules to help with the visualization of how the lipid-like densities align with the surface properties of the pore.

– We modified the text to clarify the correlation between surface properties and bilayer formation and to incorporate the description of new information included in the figures.

4) You have included new figures highlighting the tight and loose interfaces in ECD, TMD, and ICD (Figure 3 —figure supplement 1-3). This is interesting, but an accurate description of the main interactions is not found in the main text.

We expanded on the description of the intersubunit interactions refering to Figure 3—figure supplement 1-3 in the "8C-8A(IL1^25^) chimeras form heptameric channels" section in page 6.

5) You could elaborate on the description of D50/K51 rearrangements in the Result section. You state, "When K51 points toward the pore, it creates a groove between the subunits (Figure 4D)". Although this statement seems correct in the current context, the sentence confuses the overall interpretation of D50/K51 rearrangements in LRRC8C channels because the statement only applies for structures of class 1 and 3 but not for classes 2, 4, and 5. For example, the structure of class 2 only has one K51 side chain pointing toward the pore (Figure 5 – Figure sup2). Still, it keeps two loose interfaces anyway (Figure 5C), which suggests that D50/K51 rearrangement is not necessarily coupled with the loose interface as one could interpret from the text.

We modified the last paragraphs of "8C-8A(IL1^25^) chimeras form heptameric channels (page 7)" and "Structural heterogeneity of 8C-8A(IL1^25^) chimeras (page 8)" sections to clarify that the D50/K51 orientation and the presence of loose interface do exhibit variability in different classes and are not necessarily correlated.

6) Page 4, Line 80-82 "The oligomeric state directly impacts pore size, pharmacology, and permeability to large anionic solutes. Our results also suggest that associated membrane lipids may be involved in channel gating and regulation".This work did not evaluate if the oligomeric state affects the pharmacology of the channels. As discussed in the point-by-point letter, current evidence supports the hypothesis that pore size could impact pharmacology, but this could also be related to the intrinsic properties of LRRC8C. Also, in view that there is no evidence supporting the role of lipids in channel function, the above sentence is speculative. Thus, this entire sentence should be restricted to the Discussion section only.

We removed these sentences from the introduction.

7) Figure 1: Please Indicate in the figure legend what the symbols shown below the alignment in panel A (*. :) mean.

We added the description of the symbols to the figure legend.

8) Please specify in the figure legends what class is shown in figures 2, 3, and 4. Are these the class 1 structure?

We used the class 1 structure in these figures and clarified this in the figure legends.

9) Figure 4: The arrow showing rotation between structures of panels D and E seems to be in the wrong direction. Please check.

We checked the figure to make sure the arrow direction was correct.

10) Figure legend 7: "B) A sliced view of the unsharpened cryo-EM map (grey, transparent) with the ribbon representation of the 8C-8A(IL125) structure".The ribbon representation is not shown in the new version of the manuscript. Please, check.

We checked the figure and confirmed that the legend is accurate. The ribbon representation is present in the figure.

11) Table 3: The concentration unit for sucrose concentration is missing. Please check.

We added the concentration unit for sucrose.